# CAREER: A Foundation Model for Labor Sequence Data

**Keyon Vafa**                                                                    *kvafa@g.harvard.edu*
*Harvard University*

**Emil Palikot**
*Stanford University*

**Tianyu Du**
*Stanford University*

**Ayush Kanodia**
*Stanford University*

**Susan Athey**
*Stanford University*

**David M. Blei**
*Columbia University*

**Reviewed on OpenReview:** https://openreview.net/forum?id=4i1MXH8Sle

## Abstract

Labor economists regularly analyze employment data by fitting predictive models to small, carefully constructed longitudinal survey datasets. Although machine learning methods offer promise for such problems, these survey datasets are too small to take advantage of them. In recent years large datasets of online resumes have also become available, providing data about the career trajectories of millions of individuals. However, standard econometric models cannot take advantage of their scale or incorporate them into the analysis of survey data. To this end we develop CAREER, a foundation model for job sequences. CAREER is first fit to large, passively-collected resume data and then fine-tuned to smaller, better-curated datasets for economic inferences. We fit CAREER to a dataset of 24 million job sequences from resumes, and adjust it on small longitudinal survey datasets. We find that CAREER forms accurate predictions of job sequences, outperforming econometric baselines on three widely-used economics datasets. We further find that CAREER can be used to form good predictions of other downstream variables. For example, incorporating CAREER into a wage model provides better predictions than the econometric models currently in use.

## 1 Introduction

In labor economics, many analyses rely on models for predicting an individual's future occupations. These models are crucial for estimating important economic quantities, such as gender or racial differences in unemployment (Hall, 1972; Fairlie & Sundstrom, 1999); they underpin causal analyses and decompositions that rely on simulating counterfactual occupations for individuals (Brown et al., 1980; Schubert et al., 2021); and they also inform policy, by forecasting occupations with rising or declining market shares. More accurate predictive models opens the door to more reliable estimates of these important quantities.

In this paper we develop a representation-learning method—a transformer adapted for modeling jobs—for building such predictive models of occupation. Our model is first fit to large-scale, passively-collected resume

data and then fine-tuned to more curated economics datasets, which are carefully collected for unbiased generalization to the larger population. The representation it learns is effective both for predicting job trajectories and for conditioning in downstream economic analyses.

In the past, labor economics analyses only involved fitting predictive models to small datasets, specifically longitudinal surveys that follow a cohort of individuals during their working career (Panel Study of Income Dynamics, 2021; Bureau of Labor Statistics, 2019a). Such surveys have been carefully collected to represent national demographics, ensuring that the economic analyses can generalize, but they are also very small, usually containing only thousands of workers. As a consequence, prior models of occupation trajectories have been based on very simple sequential assumptions, such as where a worker's next occupation depends only on their most recent occupation (Hall, 1972) or a few summary statistics about their past (Blau & Riphahn, 1999).

In recent years, however, much larger datasets of online resumes have also become available. These datasets contain the occupation histories of millions of individuals, potentially revealing complex information and patterns about career trajectories. But, while one might hope these datasets can improve our economic analyses, there are fundamental difficulties to using them. First, they are passively collected and likely represent a biased sample of the population. Second, they are noisy, since the occupation sequences in the data are derived from text analysis of raw resumes. Finally, they generally omit important economic variables such as demographics and wage, which are essential for the kinds of quantities that economists would like to estimate.

To overcome these challenges, we develop CAREER, a machine learning model of occupation trajectories. CAREER is a foundation model (Bommasani et al., 2021): it learns an initial representation of job history from large-scale resume data that is then adjusted on downstream survey datasets. CAREER is based on the transformer architecture (Vaswani et al., 2017), which underpins foundation models in natural language processing (Devlin et al., 2019; Radford et al., 2019) and other fields, such as music generation (Copet et al., 2023) and coding (Li et al., 2023). We will show that CAREER's representations provide effective predictions of occupations on survey datasets used for economic analysis, and can be used as inputs to economic models for other downstream applications.

To study this model empirically, we pretrain CAREER on a dataset of 24 million passively-collected resumes provided by Zippia, a career planning company. We then fine-tune CAREER's representations of job sequences to make predictions on three widely-used economic datasets: the National Longitudinal Survey of Youth 1979 (NLSY79), another cohort from the same survey (NLSY97), and the Panel Study of Income Dynamics (PSID). In contrast to resume data, these well-curated datasets are representative of the larger population. It is with these survey datasets that economists make inferences, ensuring their analyses generalize. In this study, we find that CAREER outperforms standard econometric models for predicting and forecasting occupations on these survey datasets.

We further find that CAREER can be used to form good predictions of other downstream variables; incorporating CAREER into a wage model provides better predictions than the econometric models currently in use. We release code so that practitioners can train CAREER for their own problems.[1]

In summary, we demonstrate that CAREER can leverage large-scale resume data to make accurate predictions on important datasets from economics. Thus CAREER ties together economic models for understanding career trajectories with transformer-based foundation models. (See Section 3 for details of related work.)

## 2 CAREER

Given an individual's career history, what is the probability distribution of their occupation in the next timestep? We go over a class of models for predicting occupations before introducing CAREER, one such model based on transformers and transfer learning.

---

[1] https://github.com/keyonvafa/career-code

## 2.1 Occupation Models

Consider an individual worker. This person's career can be defined as a series of timesteps. Here, we use a timestep of one year. At each timestep, this individual works in a job: it could be the same job as the previous timestep, or a different job. (Note we use the terms "occupation" and "job" synonymously.) We consider "unemployed" and "out-of-labor-force" to be special types of jobs.

Define an **occupation model** to be a probability distribution over sequences of jobs. An occupation model predicts a worker's job at each timestep as a function of all previous jobs and other observed characteristics of the worker.

More formally, define an individual's career to be a sequence $(y_1, \ldots, y_T)$, where each $y_t \in \{1, \ldots, J\}$ indexes one of $J$ occupations at time $t$. Occupations are categorical; one example of a sequence could be ("cashier", "salesperson", ... , "sales manager"). At each timestep, an individual is also associated with $C$ observed covariates $x_t = \{x_{tc}\}_{c=1}^C$. Covariates are also categorical, with $x_{tc} \in \{1, \ldots, N_c\}$. For example, if $c$ corresponds to the most recent educational degree, $x_{tc}$ could be "high school diploma" or "bachelors", and $N_c$ is the number of types of educational degrees.[2] Define $\mathbf{y}_t = (y_1, \ldots, y_t)$ to index all jobs that have occurred up to time $t$, with the analogous definition for $\mathbf{x}_t$.

At each timestep, an occupation model predicts an individual's job in the next timestep, $p(y_t|\mathbf{y}_{t-1}, \mathbf{x}_t)$. This distribution conditions on covariates from the same timestep because these are "pre-transition." For example, an individual's most recent educational degree is available to the model as it predicts their next job.

Note that an occupation model is a predictive rather than structural model. The model does not incorporate unobserved characteristics, like skill, when making predictions. Instead, it implicitly marginalizes over these unobserved variables, incorporating them into its predictive distribution.

## 2.2 Representation-Based Two-Stage Models

An occupation model's predictions are governed by an individual's career history; both whether an individual changes jobs and the specific job they may transition to depend on current and previous jobs and covariates.

We consider a class of occupation models that make predictions by conditioning on a low-dimensional representation of work history, $h_t(\mathbf{y}_{t-1}, \mathbf{x}_t) \in \mathbb{R}^D$. This representation is assumed to be a sufficient statistic of the past; $h_t(\mathbf{y}_{t-1}, \mathbf{x}_t)$ should contain the relevant observed information for predicting the next job.

Since individuals frequently stay in the same job between timesteps, we propose a class of models that make predictions in two stages. These models first predict whether an individual changes jobs, after which they predict the specific job to which an individual transitions. The representation is used in both stages.

In the first stage, the career representation $h_t(\mathbf{y}_{t-1}, \mathbf{x}_t)$ is used to predict whether an individual changes jobs. Define the binary variable $s_t$ to be 1 if a worker's job at time $t$ is different from that at time $t-1$, and 0 otherwise. The first stage is modeled by

$$s_t|\mathbf{y}_{t-1}, \mathbf{x}_t \sim \text{Bernoulli}\left(\sigma(\eta \cdot h_t(\mathbf{y}_{t-1}, \mathbf{x}_t))\right), \tag{1}$$

where $\sigma(\cdot)$ is the logistic function and $\eta \in \mathbb{R}^D$ is a vector of coefficients.

If the model predicts that an individual will transition jobs, it only considers jobs that are different from the individual's most recent job. To formulate this prediction, it combines the career representation with a vector of occupation-specific coefficients $\beta_j \in \mathbb{R}^D$:

$$p(y_t = j|\mathbf{y}_{t-1}, \mathbf{x}_t, s_t = 1) = \frac{\exp\{\beta_j \cdot h_t(\mathbf{y}_{t-1}, \mathbf{x}_t)\}}{\sum_{j' \neq y_{t-1}} \exp\{\beta_{j'} \cdot h_t(\mathbf{y}_{t-1}, \mathbf{x}_t)\}}. \tag{2}$$

Otherwise, the next job is deterministic:

$$p(y_t = j|\mathbf{y}_{t-1}, \mathbf{x}_t, s_t = 0) = \delta_{j=y_{t-1}}. \tag{3}$$

---

[2]Some covariates may not evolve over time. We encode them as time-varying without loss of generality.

Two-stage prediction improves the accuracy of occupation models. Moreover, many analyses of occupational mobility focus on whether workers transition jobs rather than the specific job they transition to (Kambourov & Manovskii, 2008). By separating the mechanism by which a worker either keeps or changes jobs ($\eta$) and the specific job they may transition to ($\beta_j$), two-stage models are more interpretable for studying occupational change.

Equations 1 to 3 define a two-stage representation-based occupation model. In the next section, we introduce CAREER, one such model based on transformers.

## 2.3  CAREER Model

We develop a two-stage representation-based occupation model called **CAREER**.[3] This model uses a transformer to parameterize a representation of an individual's history. CAREER is a foundation model: it is pretrained on a large resumes dataset and fine-tuned to make predictions on small survey datasets.

**Transformers.**  A transformer is a sequence model that uses neural networks to learn representations of discrete tokens (Vaswani et al., 2017). Transformers were originally developed for natural language processing (NLP), to predict words in a sentence. Transformers are able to model complex dependencies between words, and they are a critical component of large language models (Radford et al., 2018; Devlin et al., 2019; Radford et al., 2019; Brown et al., 2020; Hoffmann et al., 2022; Touvron et al., 2023).

CAREER is an occupation model that uses a transformer to parameterize a low-dimensional representation of careers. While transformers were developed to model sequences of words, CAREER uses a transformer to model sequences of jobs. The transformer enables the model to represent complex career trajectories.

CAREER is similar to the transformers used in NLP, but with two modifications. First, as described in Section 2.2, the model makes predictions in two stages, making it better-suited to model workers who stay in the same job through consecutive timesteps. (In contrast, words seldom repeat.) Second, while language models only condition on previous words, each career is also associated with covariates $\mathbf{x}$ that may affect transition distributions (see Equation 2). We adapt the transformer to these two changes.

**Parameterization.**  CAREER's computation graph is depicted in Figure 1. Note that in this section we provide a simplified description of the ideas underlying the transformer. Appendix C contains a full description of the model.

CAREER iteratively builds a representation of career history, $h_t(\mathbf{y}_{t-1}, \mathbf{x}_t) \in \mathbb{R}^D$, using a stack of $L$ layers. Each layer applies a series of computations to the previous layer's output to produce its own layer-specific representation. The first layer's representation, $h_t^{(1)}(\mathbf{y}_{t-1}, \mathbf{x}_t)$, considers only the most recent job and covariates. At each subsequent layer $\ell$, the transformer forms a representation $h_t^{(\ell)}(\mathbf{y}_{t-1}, \mathbf{x}_t)$ by combining the representation of the most recent job with those of preceding jobs. Representations become increasingly complex at each layer, and the final layer's representation, $h_t^{(L)}(\mathbf{y}_{t-1}, \mathbf{x}_t)$, is used to make predictions following Equations 1 to 3. We drop the explicit dependence on $\mathbf{y}_{t-1}$ and $\mathbf{x}_t$ going forward, and instead denote each layer's representation as $h_t^{(\ell)}$.

The first layer's representation combines the previous job, the most recent covariates, and the position of the job in the career. It first embeds each of these variables in $D$-dimensional space. Define an embedding function for occupations, $e_y : [J] \to \mathbb{R}^D$. Additionally, define a separate embedding function for each covariate, $\{e_c\}_{c=1}^C$, with each $e_c : [N_c] \to \mathbb{R}^D$. Finally, define $e_t : [T] \to \mathbb{R}^D$ to embed the position of the sequence, where $T$ denotes the number of possible sequence lengths. The first-layer representation $h_t^{(1)}$ sums these embeddings:

$$h_t^{(1)} = e_y(y_{t-1}) + \sum_c e_c(x_{tc}) + e_t(t). \tag{4}$$

---

[3]CAREER is short for "Contextual Attention-based Representations of Employment Encoded from Resumes."

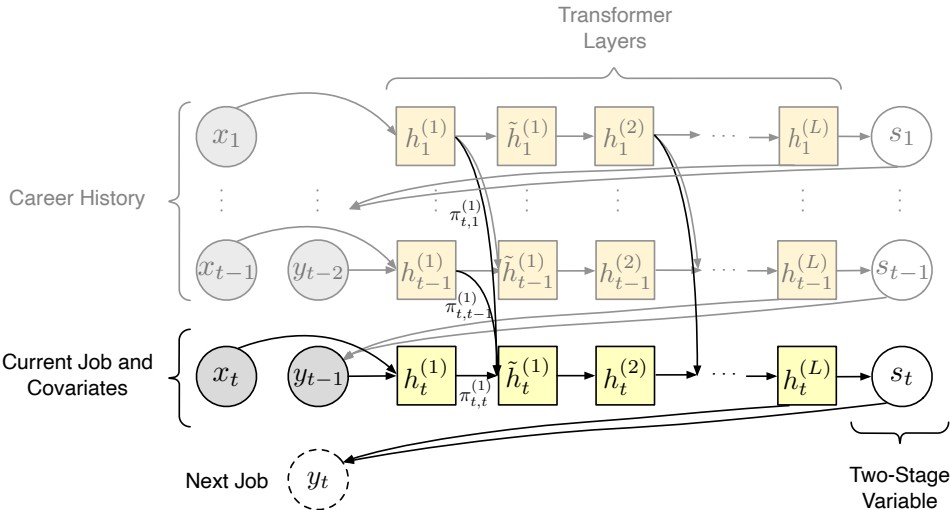

**Figure 1:** CAREER's computation graph. CAREER parameterizes a low-dimensional representation of an individual's career history with a transformer, which it uses to predict the next job.

For each subsequent layer $\ell$, the transformer combines representations of the most recent job with those of the preceding jobs and passes them through a neural network:

$$\pi_{t,t'}^{(\ell)} \propto \exp\left\{\left(h_t^{(\ell)}\right)^\top W^{(\ell)} h_{t'}^{(\ell)}\right\} \quad \text{for all } t' \leq t \tag{5}$$

$$\tilde{h}_t^{(\ell)} = h_t^{(\ell)} + \sum_{t'=1}^{t} \pi_{t,t'}^{(\ell)} * h_{t'}^{(\ell)} \tag{6}$$

$$h_t^{(\ell+1)} = \text{FFN}^{(\ell)}\left(\tilde{h}_t^{(\ell)}\right), \tag{7}$$

where $W^{(\ell)} \in \mathbb{R}^{D \times D}$ is a model parameter and $\text{FFN}^{(\ell)}$ is a two-layer feedforward neural network specific to layer $\ell$, with $\text{FFN}^{(\ell)} : \mathbb{R}^D \to \mathbb{R}^D$.

The weights $\{\pi_{t,t'}^{(\ell)}\}$ are referred to as *attention weights*, and they are determined by the career representations and $W^{(\ell)}$. The attention weights are non-negative and normalized to sum to 1. The matrix $W^{(\ell)}$ can be interpreted as a similarity matrix; if $W^{(\ell)}$ is the identity matrix, occupations $t$ and $t'$ that have similar representations will have large attention weights, and thus $t'$ would contribute more to the weighted average in Equation 6. Conversely, if $W^{(\ell)}$ is the negative identity matrix, occupations that have differing representations will have large attention weights.[4] The final computation of each layer involves passing the intermediate representation $\tilde{h}_t^{(\ell)}$ through a neural network, which ensures that representations capture complex nonlinear interactions.

The computations in Equations 5 to 7 are repeated for each of the $L$ layers. The last layer's representation is used to predict the next job:

$$p(y_t | \mathbf{y}_{t-1}, \mathbf{x}_t) = \text{two-stage-softmax}\left(h_t^{(L)}; \eta, \beta\right), \tag{8}$$

where "two-stage-softmax" refers to the operation in Equations 1 to 3, parameterized by $\eta$ and $\beta$.

All of CAREER's parameters — including the embedding functions, similarity matrices, feed-forward neural networks, and regression coefficients $\eta$ and $\beta$ — are estimated by maximizing the likelihood in Equation 8 with stochastic gradient descent (SGD), marginalizing out the latent variable $s_t$.

**Transfer learning.** Economists build occupation models using survey datasets that have been carefully collected to represent national demographics. In the United States, these datasets contain a small number

---

[4]In practice, transformers use multiple attention weights to perform *multi-headed attention* (Appendix C).

of individuals. While transformers have been successfully applied to large NLP datasets, they are prone to overfitting on small datasets (Kaplan et al., 2020; Dosovitskiy et al., 2021; Variš & Bojar, 2021). As such, CAREER may not learn useful representations solely from small survey datasets.

In recent years, however, much larger datasets of online resumes have also become available. Although these passively-collected datasets provide job sequences of many more individuals, they are not used for economic estimation for a few reasons. The occupation sequences from resumes are imputed from short textual descriptions, a process that inevitably introduces more noise and errors than collecting data from detailed questionnaires. Additionally, individuals may not accurately list their work experiences on resumes (Wexler, 2006), and important economic variables relating to demographics and wage are not available. Finally, these datasets are not constructed to ensure that they are representative of the general population.

Between these two types of data is a tension. On the one hand, resume data is large-scale and contains valuable information about employment patterns. On the other hand, survey datasets are carefully collected, designed to help make economic inferences that are robust and generalizable.

Thus CAREER incorporates the patterns embedded in large-scale resume data into the analysis of survey datasets. It does this through transfer learning: CAREER is first *pretrained* on a large dataset of resumes to learn an initial representation of careers. When CAREER is then fit to a small survey dataset, parameters are not initialized randomly; instead, they are initialized with the representations learned from resumes. After initialization, all parameters are *fine-tuned* on the small dataset by optimizing the likelihood. Because the objective function is non-convex, learned representations depend on their initial values. Initializing with the pretrained representations ensures that the model does not need to re-learn representations on the small dataset. Instead, it only adjusts representations to account for dataset differences.

CAREER is designed to be a foundation model (Bommasani et al., 2021): after pretraining on large-scale resume data, CAREER can be transferred to downstream survey datasets. CAREER takes inspiration from foundation models developed in other domains, such as NLP (Devlin et al., 2019; Radford et al., 2018; 2019; Brown et al., 2020), image generation (Ramesh et al., 2021), music generation (Copet et al., 2023), and coding (Li et al., 2023). For example, in NLP, transformers are pretrained on large corpora, such as unpublished books or web crawl data, and then fine-tuned to make predictions on small datasets such as movie reviews (Devlin et al., 2019). Our approach is analogous. Although the resumes dataset may not be representative or carefully curated, it contains many more job sequences than most survey datasets. This volume enables CAREER to learn representations that transfer to survey datasets.

## 3  Related Work

Many economic analyses use log-linear models to predict jobs in survey datasets (Boskin, 1974; Schmidt & Strauss, 1975). These models typically use small state spaces consisting of only a few occupation categories. For example, some studies categorize occupations into broad skill groups (Keane & Wolpin, 1997; Cortes, 2016); unemployment analyses only consider employment status (employed, unemployed, and out-of-labor-force) (Hall, 1972; Lauerova & Terrell, 2007); and researchers studying occupational mobility only consider occupational change, a binary variable indicating whether an individual changes jobs (Kambourov & Manovskii, 2008; Guvenen et al., 2020). Although transitions between occupations may depend richly on history, many of these models condition on only the most recent job and a few manually constructed summary statistics about history to make predictions (Hall, 1972; Blau & Riphahn, 1999). In contrast to these methods, CAREER is nonlinear and conditions on every job in an individual's history. The model learns complex representations of careers without relying on manually constructed features. Moreover, CAREER can effectively predict from among hundreds of occupations.

Recently, the proliferation of business networking platforms has resulted in the availability of large resume datasets. Schubert et al. (2021) use a large resume dataset to construct a first-order Markov model of job transitions; CAREER, which conditions on all jobs in a history, makes more accurate predictions than a Markov model. Models developed in the data mining community rely on resume-specific features such as stock prices (Xu et al., 2018), worker skill (Ghosh et al., 2020), network information (Meng et al., 2019; Zhang et al., 2021), and textual descriptions (He et al., 2021), and are not applicable to survey datasets, as is our goal in this pa-

per (other models reduce to a first-order Markov model without these features (Dave et al., 2018; Zhang et al., 2020)). The most suitable model for survey datasets from this line of work is NEMO, an LSTM-based model that is trained on large resume datasets (Li et al., 2017). In contrast to these approaches, our primary concern is not modeling resume datasets directly, but rather modeling survey datasets. Our experiments demonstrate that a transfer learning approach, from resume data to survey data, improves modeling performance.

Recent work in econometrics has applied machine learning methods to sequences of jobs and other discrete data. In the context of consumer grocery shopping behavior, Ruiz et al. (2020) and Donnelly et al. (2021) introduce a method called SHOPPER, based on matrix factorization, for creating embeddings of grocery products. SHOPPER can be thought of as a foundation model, and Donnelly et al. (2021) show that the approach of building embeddings using data from unrelated product categories improves predictions for a given category relative to traditional economic models focused only on one category, while Ruiz et al. (2020) expand the setup to predict the next product in a sequence added to a shopping basket. One of the baselines we consider in this paper is a "bag-of-jobs" model similar to SHOPPER. Like the transformer-based model, the bag-of-jobs model conditions on every job in an individual's history, but it uses relatively simple representations of careers. Our empirical studies demonstrate that CAREER learns complex representations that are better at modeling job sequences. Rajkumar et al. (2021) build on SHOPPER and propose a Bayesian factorization method for predicting job transitions. Similar to CAREER, they predict jobs in two stages. However, their method only conditions on the most recent job in an individual's history. In our empirical studies, we show that models like CAREER that condition on every job in an individual's history form more accurate predictions than Markov models.

CAREER is a foundation model based on a transformer, an architecture initially developed to represent sequences of words in natural language processing (NLP). In econometrics, transformers have been applied to the text of job descriptions to predict their salaries (Bana, 2021) or authenticity (Naudé et al., 2022); rather than modeling text, we use transformers to model sequences of occupations. Transformers underpin foundation models developed for text (Devlin et al., 2019; Radford et al., 2018; 2019; Brown et al., 2020) and other domains such as image generation (Ramesh et al., 2021), music generation (Copet et al., 2023), and coding (Li et al., 2023). Occupational trajectories have simpler structure than the data modeled in these other domains; for example, per-year career trajectories are shorter than the text sequences used to train large language models, and there are less possible occupations than words in a vocabulary. Thus, foundation models for occupations may not require as much data as those in other domains. Indeed, CAREER is smaller than foundation models in other domains and uses less data, but still forms effective predictions on downstream tasks.

Our paper also contributes to the growing field of transfer learning in economics. Most existing transfer learning approaches in the economics literature have focused on fine-tuning existing models on image or text data, where the effectiveness of transfer learning is better understood. For example, one existing line of work has fine-tune pretrained convolutional neural networks on satellite images in order to predict economic indicators (Price & Atkinson, 2022; Yeh et al., 2020; Persello & Kuffer, 2020), while another has fine-tuned BERT on financial and economic text (Araci, 2019; Bana, 2021). Our work differs from these papers in that we focus on a new domain (job sequences on survey data instead of images or text) and also identify a novel pretraining corpus that helps with this task (large-scale, passively-collected resume data).

## 4 Empirical Studies

We assess CAREER's ability to predict jobs and provide useful representations of careers. We pretrain CAREER on a large dataset of resumes, and transfer these representations to small, commonly used survey datasets. We study straightforward prediction (predicting jobs in a career with random train and test set splits) and forecasting (training on jobs before a certain year and predicting on future years). With the transferred representations, the model is better than econometric baselines at both prediction and forecasting. Additionally, we demonstrate that CAREER's representations can be incorporated into standard wage prediction models to make better predictions.

**Resume pretraining.** We pretrain CAREER on a large dataset of resumes provided by Zippia Inc., a career planning company. This dataset contains resumes from 23.7 million working Americans. Each job is encoded into one of 330 occupational codes, using the coding scheme of Autor & Dorn (2013). We transform

resumes into sequences of jobs by including an occupation's code for each year in the resume. For years with multiple jobs, we take the job the individual spent the most time in. We include three covariates: the year each job in an individual's career took place, along with the individual's state of residence and most recent educational degree. We denote missing covariates with a special token. (Appendix D contains an exploratory data analysis of the resume data.) CAREER uses a 12-layer transformer with 5.6 million parameters. Pretraining CAREER on the resumes data takes 18 hours on a single GPU. (See Appendix G for more details on the model and hyperparameters.)

**Survey datasets.** We transfer CAREER to three widely-used survey datasets: two cohorts from the National Longitudinal Survey of Youth (NLSY79 and NLSY97) and the Panel Study of Income Dynamics (PSID). These datasets have been carefully constructed to be representative of the general population, and they are widely used by economists for estimating economic quantities. NLSY79 is a longitudinal panel survey following a cohort of Americans who were between 14 and 22 when the survey began in 1979, while NLSY97 follows a different cohort of individuals who were between 12 and 17 when the survey began in 1997. PSID is a longitudinal survey following a sample of American families, with individuals added over the years. These surveys are publicly available and allow for linking individuals over time. See Appendix F for more information about how the data is formed.

Compared to the resumes dataset, these survey datasets are small: we use slices of NLSY79, NLSY97, and PSID that contain 12 thousand, 9 thousand, and 12 thousand individuals, respectively. We convert all datasets so that occupations follow the `occ1990dd` encoding (Autor & Dorn, 2013), which contains 330 discrete occupations. Both the resume dataset and survey datasets contain education, location, and year covariates, while the survey datasets additionally include demographic variables (gender and race/ethnicity). The distribution of job sequences in resumes differs in meaningful ways from those in the survey datasets; for example, manual laborers are under-represented and college graduates are over-represented in resume data. Appendix D includes more details for how these datasets differ.

We pretrain CAREER on the large resumes dataset and fine-tune on the smaller survey datasets. We fine-tune by initializing each model's weights with those of the pretrained model. The fine-tuning process is efficient; although CAREER has 5.6 million parameters, fine-tuning on one GPU takes 13 minutes on NLSY79, 7 minutes on NLSY97, and 23 minutes on PSID.

We compare CAREER to several baseline models: a second-order linear regression with covariates and hand-constructed summary statistics about past employment; a bag-of-jobs model inspired by SHOPPER (Ruiz et al., 2020) that conditions on all jobs and covariates in a history but combines representations linearly; and several baselines developed in the data-mining community for modeling worker profiles: NEMO (Li et al., 2017), job representation learning (Dave et al., 2018), and Job2Vec (Zhang et al., 2020). As described in Section 3, the baselines developed in the data-mining community for modeling worker profiles cannot be applied directly to economic survey datasets and thus require modifications, described in detail in Appendix G. These baselines are all trained on survey data. We also compare to two additional versions of CAREER — one without pretraining or two-stage prediction, the other only without two-stage prediction — to assess the sources of CAREER's improvements. All models use the covariates we included for resume pretraining, in addition to demographic covariates (which are recorded for the survey datasets but are unavailable for resumes).

We divide all survey datasets into 70/10/20 train/validation/test splits, and train all models by optimizing the log-likelihood with Adam (Kingma & Ba, 2015). We evaluate the predictive performance of each model by computing held-out perplexity, a common metric in NLP for evaluating probabilistic sequence models. The perplexity of a sequence model $p$ on a sequence $y_1, \ldots, y_T$ is $\exp\{-\frac{1}{T}\sum_{t=1}^{T}\log p(y_t|\mathbf{y}_{t-1}, \mathbf{x}_t)\}$. It is a monotonic transformation of log-likelihood; better predictive models have lower perplexities. We train all models to convergence and use the checkpoint with the best validation perplexity. Detailed experimental settings and training details are described in Appendix G.

Figure 2a compares the test-set perplexity of each model. With the transferred representations, CAREER makes the best predictions on all survey datasets. The baselines developed in the data mining literature, which were designed to model large resume datasets while relying on resume-specific features, struggle to make good predictions on these small survey datasets, performing on par with standard econometric baselines.

|  | PSID | NLSY79 | NLSY97 |
|---|---|---|---|
| Markov regression (Hall, 1972) | 18.97 ±0.10 | 15.03 ±0.03 | 20.81 ±0.02 |
| NEMO (Li et al., 2017) | 17.58 ±0.04 | 12.82 ±0.04 | 18.38 ±0.08 |
| Job rep. learning (Dave et al., 2018) | 17.23 ±0.16 | 14.71 ±0.02 | 16.83 ±0.03 |
| Job2Vec (Zhang et al., 2020) | 16.48 ±0.13 | 14.46 ±0.01 | 16.20 ±0.02 |
| Bag-of-jobs (Ruiz et al., 2020) | 16.21 ±0.08 | 13.09 ±0.03 | 16.20 ±0.01 |
| CAREER (vanilla) | 15.26 ±0.08 | 12.20 ±0.04 | 16.19 ±0.04 |
| CAREER (two-stage) | 14.79 ±0.04 | 12.00 ±0.00 | 15.22 ±0.03 |
| CAREER (two-stage + pretrain) | **13.88 ±0.01** | **11.32 ±0.00** | **14.15 ±0.03** |

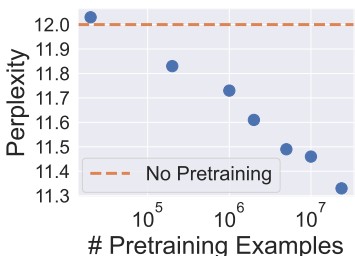

**(a)** Test perplexity on survey datasets (lower is better). Results are averaged over three random seeds. CAREER (vanilla) includes covariates but not two-stage prediction or pretraining; CAREER (two-stage) adds two-stage prediction.

**(b)** CAREER's scaling law on NLSY79 as a function of pretraining data volume.

**Figure 2:** Prediction results on longitudinal survey datasets and scaling law.

Pretraining is the biggest source of CAREER's improvements. Although the resume data is noisy and differs in many ways from the survey datasets used for economic prediction, CAREER learns representations of work experiences that aid its predictive performance. In Appendix E we show that modifying the baselines to incorporate two-stage prediction (Equations 1 to 3) improves their performance, although CAREER still makes the best predictions across datasets. We include further qualitative analysis of CAREER's predictions in Appendix B.

To assess how the volume of resumes used for pretraining affects CAREER's predictions on survey datasets, we downsample the resume dataset and transfer to survey datasets. The scaling law for NLSY79 is depicted in Figure 2b. When there are less than 20,000 examples in the resume dataset, pretraining CAREER does not offer any improvement. The relationship between pretraining volume and fine-tuned perplexity follows a power law, similar to scaling laws in NLP (Kaplan et al., 2020).

We also assess CAREER's ability to forecast future career trajectories. In contrast to predicting held-out sequences, forecasting involves training models on all sequences before a specific year. To predict future jobs for an individual, the fitted model is used to estimate job probabilities six years into the future by sampling multi-year trajectories. This setting is useful for assessing a model's ability to make long-term predictions, especially as occupational trends change over time.

We evaluate CAREER's forecasting abilities on NLSY97 and PSID. (These datasets are more valuable for forecasting than NLSY79, which follows a cohort that is near or past retirement age.) We train models on all sequences (holding out 10% as a validation set), without including any observations after 2014. When pretraining CAREER on resumes, we also make sure to only include examples up to 2014. Table 1 and Table 2 compare the forecasting performance of all models (results are averaged over three random seeds). CAREER makes the best overall forecasts. CAREER has a significant advantage over baselines at making long-term forecasts, yielding a 17% advantage over the best baseline for 6-year forecasts on NLSY97. Again, the baselines developed for resume data mining, which had been developed to model much larger corpora, struggle to make good predictions on these smaller survey datasets.

**Using CAREER's representations to predict wage.** In addition to forming job predictions, CAREER learns low-dimensional representations of job histories. These representations can be used as inputs to economic models for downstream applications.

As an example of how CAREER's representations can be incorporated into other economic models, we use CAREER to predict wages. Economists build wage prediction models in order to estimate important economic quantities, such as the adjusted gender wage gap (Blau & Kahn, 2017), wage effects in labor market discrimination (Neal & Johnson, 1996), and union wage premiums (Linneman et al., 1990). For example, to estimate the adjusted gender wage gap, Blau & Kahn (2017) regress an individual's log-wage on observable characteristics such as education, demographics, and current occupation on PSID. Rather than including

|  | Overall | 2-Year | 4-Year | 6-Year |
|---|---|---|---|---|
| Markov regression | $23.11 \pm 0.03$ | $12.50 \pm 0.00$ | $25.88 \pm 0.04$ | $36.59 \pm 0.07$ |
| Bag-of-jobs | $22.51 \pm 0.03$ | $11.98 \pm 0.00$ | $25.11 \pm 0.04$ | $36.29 \pm 0.10$ |
| NEMO | $25.26 \pm 0.02$ | $12.59 \pm 0.00$ | $28.35 \pm 0.05$ | $43.01 \pm 0.05$ |
| CAREER | $\mathbf{19.41 \pm 0.04}$ | $\mathbf{10.78 \pm 0.00}$ | $\mathbf{21.57 \pm 0.07}$ | $\mathbf{30.19 \pm 0.06}$ |

**Table 1:** Forecasting perplexity (lower is better) on NLSY97.

|  | Overall | 2-Year | 4-Year | 6-Year |
|---|---|---|---|---|
| Markov regression | $19.43 \pm 0.01$ | $11.83 \pm 0.00$ | $21.66 \pm 0.03$ | $27.89 \pm 0.01$ |
| Bag-of-jobs | $19.28 \pm 0.01$ | $11.44 \pm 0.00$ | $21.68 \pm 0.01$ | $28.14 \pm 0.05$ |
| NEMO | $18.58 \pm 0.01$ | $11.08 \pm 0.00$ | $20.67 \pm 0.03$ | $27.29 \pm 0.04$ |
| CAREER | $\mathbf{16.51 \pm 0.01}$ | $\mathbf{10.35 \pm 0.00}$ | $\mathbf{18.30 \pm 0.01}$ | $\mathbf{23.18 \pm 0.03}$ |

**Table 2:** Forecasting perplexity (lower is better) on PSID.

the full, high-dimensional job-history, the model summarizes an individual's career with summary statistics such as full-time and part-time years of experience (and their squares).

We incorporate CAREER's representation into the wage regression by adding the fitted representation for an individual's job history, $\hat{h}_i$. For log-wage $w_i$ and observed covariates $x_i$, we regress

$$w_i \sim \alpha + \theta^\top x_i + \gamma^\top \hat{h}_i, \tag{9}$$

where $\alpha$, $\theta$, and $\gamma$ are regression coefficients. CAREER's representations are trained to model occupation sequences on 80% of the samples on PSID, and are then frozen on the held-out samples; this ensures that the representations do not encode an individual's future occupations. To estimate regression coefficients, we again split the sample into five splits and evaluate $R^2$ on each held-out split, averaging over splits. We estimate standard errors over three random training seeds.

Table 3 shows that adding CAREER's representations improves wage predictions for each year. Although these representations are trained to model occupational sequences and are not adjusted to account for wage, they contain information that is predictive of wage. Moreover, pretraining is important to CAREER's predictive advantage. That models with these representations outperform simpler models in predictive performance, both for occupation modeling (the objective the representations are trained to optimize) and wage prediction (which the representations are not trained to optimize), underscores the complexity of the learned representations.

## 5 Conclusion

We introduced CAREER, a foundation model for occupational sequences. CAREER is first trained on large-scale resume data and then fine-tuned on smaller datasets of interest. When we fine-tuned CAREER to common survey datasets in labor economics, it outperformed econometric baselines in predicting and forecasting career outcomes. We demonstrated that CAREER's representations could be incorporated into wage prediction models, outperforming standard econometric models.

|  | Overall | 1990-1999 | 2000 - 2009 | 2010 - 2019 |
|---|---|---|---|---|
| Full specification from Blau & Kahn (2017) | $0.431 \pm 0.000$ | $0.424 \pm 0.000$ | $0.392 \pm 0.000$ | $0.459 \pm 0.000$ |
| Full specification + CAREER (not pretrained) | $0.493 \pm 0.000$ | $0.498 \pm 0.000$ | $0.453 \pm 0.000$ | $0.517 \pm 0.000$ |
| Full specification + CAREER (pretrained) | $\mathbf{0.504 \pm 0.001}$ | $\mathbf{0.504 \pm 0.001}$ | $\mathbf{0.461 \pm 0.000}$ | $\mathbf{0.532 \pm 0.000}$ |

**Table 3:** Held-out $R^2$ (higher is better) for wage regressions.

One direction of future research is to incorporate CAREER's representations of job history into methods for estimating adjusted quantities, like wage gaps. Underlying these methods are models that predict economic outcomes as a function of observed covariates. However, if relevant variables are omitted, the adjusted estimates may be affected; e.g., excluding work experience from wage prediction may change the magnitude of the estimated gap. In practice, economists include hand-designed summary statistics to overcome this problem, such as in Blau & Kahn (2017). CAREER provides a data-driven way to incorporate such variables—its representations of job history could be incorporated into downstream prediction models and lead to more accurate adjustments of economic quantities.

### Broader Impact Statement

As discussed, passively-collected resume datasets are not curated to represent national demographics. Pretraining CAREER on these datasets may result in representations that are affected by sampling bias. Although these representations are fine-tuned on survey datasets that are carefully constructed to represent national demographics, the biases from pretraining may propagate through fine-tuning (Ravfogel et al., 2020; Jin et al., 2021). Moreover, even in representative datasets, models may form more accurate predictions for majority groups due to data volume (Dwork et al., 2018). Thus, we encourage practitioners to audit noisy resume data, re-weight samples as necessary (Kalton, 1983), and review accuracy within demographics before using the model for downstream economic analysis.

Although resume datasets may contain personally identifiable information, all personally identifiable information had been removed before we were given access to the resume dataset we used for pretraining. Additionally, none of the longitudinal survey datasets contain personally identifiable information.

### Acknowledgements

This work is supported by NSF grant IIS 2127869, ONR grants N00014-17-1-2131 and N00014-15-1-2209, the Simons Foundation, Open Philanthropy, and the Golub Capital Social Impact Lab. Keyon Vafa is supported by the Harvard Data Science Initiative Postdoctoral Fellowship. We thank Elliott Ash, Sune Lehmann, Suresh Naidu, Matthew Salganik, and the reviewers for their thoughtful comments and suggestions. We thank Zippia for generously sharing the resume dataset. We also thank the Stanford Institute for Human-Centered Artificial Intelligence. Finally, we thank Lilia Chang, Karthik Rajkumar, and Lisa Simon upon whose research we build in this project, and especially Lisa Simon who helped obtain the data and encourage this line of research.

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

## A   Econometric Baselines

In this section, we describe baseline occupation models that economists have used to model jobs and other discrete sequences.

**Markov models and regression.**   A first-order Markov model assumes the job at each timestep depends on only the previous job (Hall, 1972; Poterba & Summers, 1986). Without covariates, a Markov model takes the form $p(y_t = j|\mathbf{y}_{t-1}) = p(y_t = j|y_{t-1})$. The optimal transition probabilities reflect the overall frequencies of individuals transitioning from occupation $y_{t-1}$ to occupation $j$. In a second-order Markov model, the next job depends on the previous two.

A multinomial logistic regression can be used to incorporate covariates:

$$p(y_t = j|\mathbf{y}_{t-1}, \mathbf{x}_t) \propto \exp\left\{\beta_j^{(0)} + \beta_j^{(1)} \cdot y_{t-1} + \sum_c \beta_j^{(c)} \cdot x_{tc}\right\}, \tag{10}$$

where $\beta_j^{(0)}$ is an occupation-specific intercept and $y_{t-1}$ and $x_{tc}$ denote $J$- and $N_c$-dimensional indicator vectors, respectively. Equation 10 depends on history only through the most recent job, although the covariates can also include hand-crafted summary statistics about the past, such as the duration of the most recent job (McCall, 1990). This model is fit by maximizing the likelihood with gradient-based methods.

**Bag-of-jobs.**   A weakness of the first-order Markov model is that it only uses the most recent job to make predictions. However, one's working history beyond the last job may inform future transitions (Blau & Riphahn, 1999; Neal, 1999).

Another baseline we consider is a *bag-of-jobs* model, inspired by SHOPPER, a probabilistic model of consumer choice (Ruiz et al., 2020). Unlike the Markov and regression models, the bag-of-jobs model conditions on every job in an individual's history. It does so by learning a low-dimensional representation of an individual's history. This model learns a unique embedding for each occupation, similar to a word embedding (Bengio et al., 2003; Mikolov et al., 2013); unlike CAREER, which learns complicated nonlinear interactions between jobs in a history, the bag-of-jobs model combines jobs into a single representation by averaging their embeddings.

The bag-of-jobs model assumes that job transitions depend on two terms: a term that captures the effect of the most recent job, and a term that captures the effect of all prior jobs. Accordingly, the model learns two types of representations: an embedding $\alpha_j \in \mathbb{R}^D$ of the most recent job $j$, and an embedding $\rho_{j'} \in \mathbb{R}^D$ for prior jobs $j'$. To combine the representations for all prior jobs into a single term, the model averages embeddings:

$$p(y_t = j|\mathbf{y}_{t-1}) \propto \exp\left\{\beta_j^{(1)} \cdot \alpha_{y_{t-1}} + \beta_j^{(2)} \cdot \left(\tfrac{1}{t-2} \sum_{t'=1}^{t-2} \rho_{y_{t'}}\right)\right\}. \tag{11}$$

Covariates can be added to the model analogously; for a single covariate, its most recent value is embedded and summed with the average embeddings for its prior values. All parameters are estimated by maximizing the likelihood in Equation 11 with SGD.

## B   Qualitative Analysis

**Rationalizing predictions.**   Figure 3 shows an example of a held-out career sequence from PSID. CAREER is much likelier than a regression and bag-of-jobs baseline to predict this individual's next job, biological

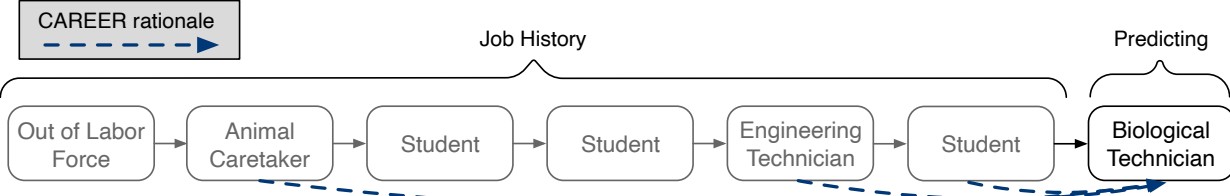

**Figure 3:** An example of a held-out job sequence on PSID along with CAREER's rationale. CAREER ranks the true next job (biological technician) as the most likely possible transition for this individual; in contrast, the regression and bag-of-jobs model rank it as 40th and 37th most likely, respectively. The rationale depicts the jobs in the history that were sufficient for CAREER's prediction.

| Model | Consecutive Repeat | Non-Consecutive Repeat | New Job |
|---|---|---|---|
| Markov regression (Hall, 1972) | 1.92 | 29.59 | 115.7 |
| NEMO (Li et al., 2017) | 1.99 | 30.84 | 105.0 |
| Job rep. learning (Dave et al., 2018) | 1.91 | 48.72 | 115.1 |
| Job2Vec (Zhang et al., 2020) | 1.88 | 45.44 | 115.1 |
| Bag-of-jobs (Ruiz et al., 2020) | 1.90 | 35.55 | 112.9 |
| CAREER | **1.87** | **12.90** | **100.1** |

**Table 4:** Held-out perplexity for different transition types on PSID (lower is better). CAREER's predictions are best across all transition types, but its biggest improvements come for predicting jobs that repeat in a career at non-consecutive timesteps..

technician. To understand CAREER's prediction, we show the model's *rationale*, or the jobs in this individual's history that are sufficient for explaining the model's prediction. (We adapt the greedy rationalization method from Vafa et al. (2021); refer to Appendix G for more details.) In this example, CAREER only needs three previous jobs to predict biological technician: animal caretaker, engineering technician, and student. The model can combine latent attributes of each job to predict the individual's next job.

**Representation similarity.** To demonstrate the quality of the learned representations, we use CAREER's fine-tuned representations on NLSY97 to find pairs of individuals with the most similar career trajectories. Specifically, we compute CAREEER's representation $h_t(\mathbf{y}_{t-1}, \mathbf{x}_t)$ for each individual in NLSY97 who has worked for four years. We then measure the similarity between all pairs by computing the cosine similarity between representations. In order to depict meaningful matches, we only consider pairs of individuals with no overlapping jobs in their histories (otherwise the model would find individuals with the exact same career trajectories). Figure 5 depicts the career histories with the most similar CAREER representations. Although none of these pairs have overlapping jobs, the model learns representations that can identify similar careers.

**Predictive improvements.** To assess the source of CAREER's predictive improvement, we decompose predictions into three categories: consecutive repeats (when the next job is the same as the previous year's), non-consecutive repeats (when the next job is different from the previous year's, but is the same as one of the prior jobs in the career), and new jobs. Table 4 shows the results on PSID. CAREER has an advantage over baselines in all three categories, but the biggest improvement comes when predicting jobs that have been repeated non-consecutively. This advantage makes sense: although consecutive repeated jobs are more common in work trajectories, CAREER's predictive advantage reveals that workers returning to jobs they previously held in their careers are also common and should be incorporated into model predictions.

We also break down predictive advantage as a function of career length for NLSY79 (Figure 4). We find that CAREER is not at an advantage when predicting an individual's first job, which makes sense since there is no additional job history to condition on. We find that for all other years of history, CAREER has a predictive advantage over the econometric baselines. However, the size of this advantage is most pronounced mid-career. For both CAREER and the baseline regression, it becomes easier to predict an individual's next

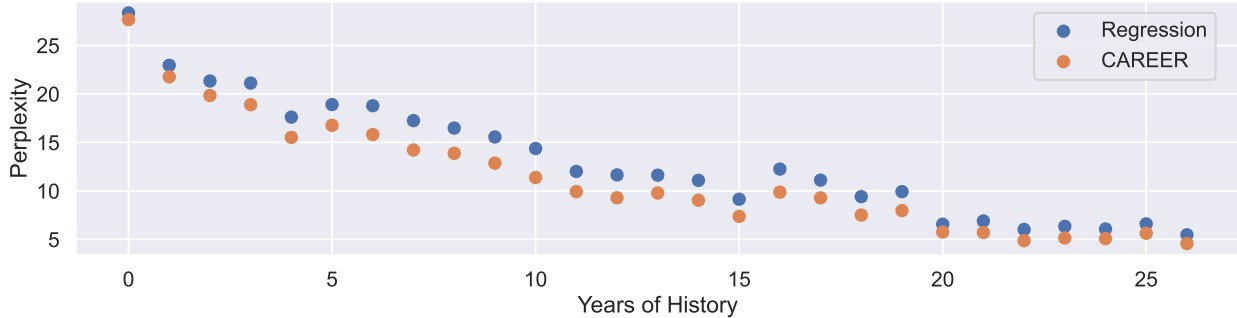

**Figure 4:** Held-out perplexity as a function of career length on NLSY79. CAREER has a predictive advantage at all points of an individual's career. The magnitude of this advantage is largest mid-career.

| Model | PSID | NLSY79 | NLSY97 |
|---|---|---|---|
| Markov regression (Hall, 1972) | $0.701 \pm 0.001$ | $0.732 \pm 0.000$ | $0.680 \pm 0.001$ |
| NEMO (Li et al., 2017) | $0.706 \pm 0.004$ | $0.731 \pm 0.002$ | $0.664 \pm 0.001$ |
| Job rep. learning (Dave et al., 2018) | $0.679 \pm 0.003$ | $0.704 \pm 0.000$ | $0.651 \pm 0.001$ |
| Job2Vec (Zhang et al., 2020) | $0.681 \pm 0.005$ | $0.705 \pm 0.000$ | $0.651 \pm 0.001$ |
| Bag-of-jobs (Ruiz et al., 2020) | $0.690 \pm 0.002$ | $0.706 \pm 0.000$ | $0.661 \pm 0.001$ |
| CAREER | $\mathbf{0.741 \pm 0.001}$ | $\mathbf{0.761 \pm 0.000}$ | $\mathbf{0.691 \pm 0.001}$ |

**Table 5:** AUC for predicting whether or not an individual transitions or keeps occupations each timestep, treating the outcome as a binary variable.

job as their career progresses, partially because individuals are more likely to stay at the same job later in their career rather than earlier. The countervailing effect is that, all else equal, CAREER's predictive advantage should be more pronounced when there are more jobs to condition on. Thus, there is a sweet spot for CAREER's advantage, between 5-10 years in an individual's career.

Table 5 compares model performances for the specific prediction of whether or not an individual changes occupations. Here, the outcome is a binary variable, indicating whether or not an individual transitioned occupations, while each model forms a prediction of an individual staying in the same occupation or leaving (summed over all possible transitions). CAREER has the highest AUC for predicting these transitions.

## C   Transformer Details

In this section, we expand on the simplified description of transformers in Section 2.3 and describe CAREER in full detail. Recall that the model estimates representations in $L$ layers, $h_t^{(1)}(\mathbf{y}_{t-1}, \mathbf{x}_t), \ldots, h_t^{(L)}(\mathbf{y}_{t-1}, \mathbf{x}_t)$,

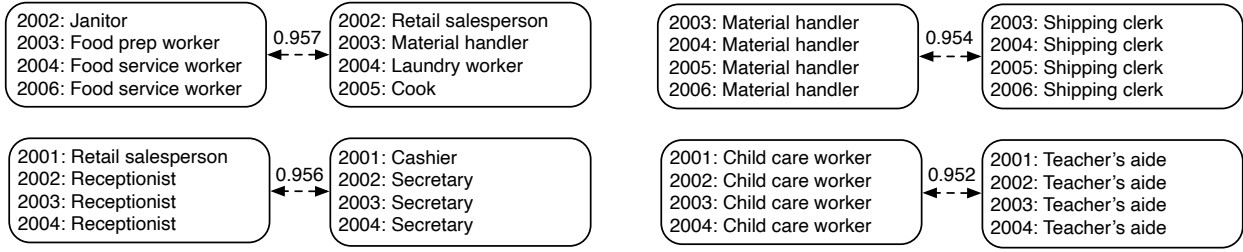

**Figure 5:** The work experiences with the most similar CAREER representations (measured with cosine similarity) for individuals with no overlapping jobs in NLSY97.

with each representation $h_t^{(\ell)} \in \mathbb{R}^D$. The final representation $h_t^{(L)}(\mathbf{y}_{t-1}, \mathbf{x}_t)$ is used to represent careers. We drop the explicit dependence on $\mathbf{y}_{t-1}$ and $\mathbf{x}_t$, and instead denote each representation as $h_t^{(\ell)}$.

The first transformer layer combines the previous occupation, the most recent covariates, and the position of the occupation in the career. It first embeds each of these variables in $D$-dimensional space. Define an embedding function for occupations, $e_y : [J] \to \mathbb{R}^D$. Additionally, define a separate embedding function for each covariate, $\{e_c\}_{c=1}^C$, with each $e_c : [N_c] \to \mathbb{R}^D$. Finally, define $e_t : [T] \to \mathbb{R}^D$ to embed the position of the sequence, where $T$ denotes the number of possible sequence lengths. The first-layer representation $h_t^{(1)}$ sums these embeddings:

$$h_t^{(1)} = e_y(y_{t-1}) + \sum_c e_c(x_{tc}) + e_t(t).  \tag{12}$$

The occupation- and covariate-specific embeddings, $e_y$ and $\{e_c\}$, are model parameters; the positional embeddings, $e_t$, are set in advance to follow a sinusoidal pattern (Vaswani et al., 2017). While these embeddings could also be parameterized, in practice the performance is similar, and using sinusoidal embeddings allows the model to generalize to career sequence lengths unseen in the training data.

At each subsequent layer, the transformer combines the representations of all occupations in a history. It combines representations by performing *multi-headed attention*, which is similar to the process described in Section 2.3 albeit with multiple attention weights per layer.

Specifically, it uses $A$ specific attention weights, or *heads*, per layer. The number of heads $A$ should be less than the representation dimension $D$. (Using $A = 1$ attention head reduces to the process described in Equations 5 and 6.) The representation dimension $D$ should be divisible by $A$; denote $K = D/A$. First, $A$ different sets of attention weights are computed:

$$
\begin{aligned}
z_{a,t,t'}^{(\ell)} &= \left(h_t^{(\ell)}\right)^\top W_a^{(\ell)} h_{t'}^{(\ell)} \quad \text{for } t' \le t \\
\pi_{a,t,t'} &= \frac{\exp\{z_{a,t,t'}\}}{\sum_k \exp\{z_{a,t,k}\}},
\end{aligned}
\tag{13}
$$

where $W_a^{(\ell)} \in \mathbb{R}^{D \times D}$ is a model parameter, specific to attention head $a$ and layer $l$.[5] Each attention head forms a convex combination with all previous representations; to differentiate between attention heads, each representation is transformed by a linear transformation $V_a^{(\ell)} \in \mathbb{R}^{K \times D}$ unique to an attention head, forming $b_{a,t}^{(\ell)} \in \mathbb{R}^K$:

$$b_{a,t}^{(\ell)} = \sum_{t'=1}^t \pi_{a,t,t'}^{(\ell)} \left(V_a^{(\ell)} h_{t'}^{(\ell)}\right).  \tag{14}$$

All attention heads are combined into a single representation by concatenating them into a single vector $g_t^{(\ell)} \in \mathbb{R}^D$:

$$g_t^{(\ell)} = \left(b_{1,t}^{(\ell)}, b_{2,t}^{(\ell)}, \ldots, b_{A,t}^{(\ell)}\right).  \tag{15}$$

To complete the multi-head attention step and form the intermediate representation $\tilde{h}_t^{(\ell)}$, the concatenated representations $g_t^{(\ell)}$ undergo a linear transformation and are summed with the pre-attention representation $h_t^{(\ell)}$:

$$\tilde{h}_t^{(\ell)} = h_t^{(\ell)} + M^{(\ell)} g_t^{(\ell)},  \tag{16}$$

with $M^{(\ell)} \in \mathbb{R}^{D \times D}$.

The intermediate representations $\tilde{h}_t^{(\ell)} \in \mathbb{R}^D$ combine the representation at timestep $t$ with those preceding timestep $t$. Each layer of the transformer concludes by taking a non-linear transformation of the intermediate

---

[5]For computational reasons, $W_a^{(\ell)}$ is decomposed into two matrices and scaled by a constant, $W_a^{(\ell)} = \frac{Q_a^{(\ell)} \left(K_a^{(\ell)}\right)^\top}{\sqrt{K}}$, with $Q_a^{(\ell)}, K_a^{(\ell)} \in \mathbb{R}^{D \times K}$.

| | | |
|---|---|---:|
| **General** | Number of individuals | 23,731,674 |
| | Number of tokens | 245,439,865 |
| | Median year | 2007 |
| **Geography** | Percent Northeast | 17.6 |
| | Percent Northcentral | 20.7 |
| | Percent South | 39.9 |
| | Percent West | 19.4 |
| | Percent without location | 2.4 |
| **Education** | Percent high school diploma | 7.2 |
| | Percent associate degree | 8.6 |
| | Percent bachelor degree | 23.1 |
| | Percent graduate degree | 4.5 |
| | Percent empty | 52.8 |
| **Broad Occupation Groups** | Percent managerial/professional specialty | 38.4 |
| | Percent technical/sales/administrative support | 34.2 |
| | Percent service | 12.0 |
| | Percent precision production/craft/repair | 7.9 |
| | Percent operator/fabricator/laborer | 7.2 |

**Table 6:** Exploratory data analysis of the resume dataset used for pretraining CAREER.

representations. This non-linear transformation does not depend on any previous representation; it only transforms $\tilde{h}_t^{(\ell)}$. Specifically, $\tilde{h}_t^{(\ell)}$ is passed through a neural network:

$$h_t^{(\ell+1)} = \tilde{h}_t^{(\ell)} + \text{FFN}^{(\ell)}\left(\tilde{h}_t^{(\ell)}\right), \tag{17}$$

where $\text{FFN}^{(\ell)}$ denotes a two-layer feedforward neural network with $N$ hidden units, with $\text{FFN}^{(\ell)} : \mathbb{R}^D \to \mathbb{R}^D$.

We repeat the multi-head attention and feedforward neural network updates above for $L$ layers, using parameters unique to each layer. We represent careers with the last-layer representation, $h_t(\mathbf{y}_{t-1}, \mathbf{x}_t) = h_t^{(L)}(\mathbf{y}_{t-1}, \mathbf{x}_t)$.

By marginalizing the two-stage prediction indicator, the conditional likelihood has the form:

$$p(y_t = j | \mathbf{y}_{t-1}, \mathbf{x}_t) = \begin{cases} \sigma(\eta \cdot h_t(\mathbf{y}_{t-1}, \mathbf{x}_t)) & \text{if } y_t = y_{t-1} \\ (1 - \sigma(\eta \cdot h_t(\mathbf{y}_{t-1}, \mathbf{x}_t))) * \frac{\exp\{\beta_j \cdot h_t(\mathbf{y}_{t-1}, \mathbf{x}_t)\}}{\sum_{j' \neq y_{t-1}} \exp\{\beta_{j'} \cdot h_t(\mathbf{y}_{t-1}, \mathbf{x}_t)\}} & \text{if } y_t \neq y_{t-1}. \end{cases} \tag{18}$$

The loss function for an individual sequence $(\mathbf{y}_t, \mathbf{x}_t)$ is given by

$$l(\mathbf{y}_T, \mathbf{x}_T) = -\sum_{t=1}^{T} \log p(y_t = j | \mathbf{y}_{t-1}, \mathbf{x}_t). \tag{19}$$

## D  Exploratory Data Analysis

Table 6 depicts summary statistics of the resume dataset provided by Zippia that is used for pretraining CAREER. Table 7 compares this resume dataset with the longitudinal survey datasets of interest.

## E  One-Stage vs Two-Stage Prediction

Table 8 compares the predictive performance of baseline occupation models when they are modified to make predictions in two stages, following Equations 1 to 3. Incorporating two-stage prediction improves

|                                                  | Resumes    | NLSY79 | NLSY97 | PSID   |
| ------------------------------------------------ | ---------- | ------ | ------ | ------ |
| Number of individuals                            | 23,731,674 | 12,270 | 8,770  | 12,338 |
| Average sequence length                          | 10.3       | 19.5   | 13.0   | 5.1    |
| Unemployed/out-of-labor-force/student available  | No         | Yes    | Yes    | Yes    |
| Median year                                      | 2007       | 1991   | 2007   | 2011   |
| Percent manual laborers                          | 7%         | 17%    | 13%    | 12%    |
| Percent college graduates                        | 56%        | 23%    | 29%    | 28%    |
| Demographic covariates available                 | No         | Yes    | Yes    | Yes    |

**Table 7:** Comparing the resume dataset used for pretraining with the three longitudinal survey datasets of interest.

|                               | PSID                        | NLSY79                      | NLSY97                      |
| ----------------------------- | --------------------------- | --------------------------- | --------------------------- |
| Markov regression (two-stage) | $15.60 \pm 0.03$            | $13.30 \pm 0.02$            | $15.47 \pm 0.00$            |
| NEMO (two-stage)              | $15.23 \pm 0.08$            | $12.37 \pm 0.04$            | $15.13 \pm 0.03$            |
| Job rep. learning (two-stage) | $15.98 \pm 0.06$            | $13.97 \pm 0.03$            | $15.43 \pm 0.01$            |
| Job2Vec (two-stage)           | $15.80 \pm 0.04$            | $13.91 \pm 0.01$            | $15.31 \pm 0.02$            |
| Bag-of-jobs (two-stage)       | $15.40 \pm 0.05$            | $12.68 \pm 0.01$            | $15.11 \pm 0.02$            |
| CAREER                        | $\mathbf{13.88 \pm 0.01}$   | $\mathbf{11.32 \pm 0.00}$   | $\mathbf{14.15 \pm 0.03}$   |

**Table 8:** Perplexity of economic baselines when they are modified to make predictions in two stages.

the performance of these baseline models compared to Figure 2a; however, CAREER still makes the best predictions on all survey datasets. For the CAREER model that was pretrained, we found that one- and two-stage models made similar predictions, suggesting that pretraining helps learn occupational mobility patterns.

## F    Data Preprocessing

In this section, we go over the data preprocessing steps we took for each dataset.

**Resumes.**    We were given access to a large dataset of resumes of American workers by Zippia, a career planning company. This dataset coded each occupation into one of 1,073 O*NET 2010 Standard Occupational Classification (SOC) categories based on the provided job titles and descriptions in resumes. We dropped all examples with missing SOC codes.

Each resume in the dataset we were given contained covariates that had been imputed based off other data in the resume. We considered three covariates: year, most recent educational degree, and location. Education degrees had been encoded into one of eight categories: high school diploma, associate, bachelors, masters, doctorate, certificate, license, and diploma. Location had been encoded into one of 50 states plus Puerto Rico, Washington D.C., and unknown, for when location could not be imputed. Some covariates also had missing entries. When an occupation's year was missing, we had to drop it from the dataset, because we could not position it in an individual's career. Whenever another covariate was missing, we replaced it with a special "missing" token. All personally identifiable information had been removed from the dataset.

We transformed each resume in the dataset into a sequence of occupations. We included an entry for each year starting from the first year an individual worked to their last year. We included a special "beginning of sequence" token to indicate when each individual's sequence started. For each year between an individual's first and last year, we added the occupation they worked in during that year. If an individual worked in multiple occupations in a year, we took the one where the individual spent more time in that year; if they were both the same amount of time in the particular year, we broke ties by adding the occupation that had started earlier in the career. For the experiments predicting future jobs directly on resumes, we added a

"no-observed-occupation" token for years where the resume did not list any occupations (we dropped this token when pretraining). Each occupation was associated with the individual's most recent educational degree, which we treated as a dynamic covariate. The year an occupation took place was also considered a dynamic categorical covariate. We treated location as static. In total, this preprocessing left us with a dataset of 23.7 million resumes, and 245 million individual occupations.

In order to transfer representations, we had to slightly modify the resumes dataset for pretraining to encode occupations and covariates into a format compatible with the survey datasets. The survey datasets we used were encoded with the "occ1990dd" occupation code (Autor & Dorn, 2013) rather than with O*NET's SOC codes, so we converted the SOC codes to occ1990dd codes using a crosswalk posted online by Destin Royer. Even after we manually added a few missing entries to the crosswalks, there were some SOC codes that did not have corresponding occ1990dd's. We gave these tokens special codes that were not used when fine-tuning on the survey datasets (because they did not correspond to occ1990dd occupations). When an individual did not work for a given year, the survey datasets differentiated between three possible states: unemployed, out-of-labor-force, and in-school. The resumes dataset did not have these categories. Thus, we initialized parameters for these three new occupational states randomly. Additionally, we did not include the "no-observed-occupation" token when pretraining, and instead dropped missing years from the sequence. Since we did not use gender and race/ethnicity covariates when pretraining, we also initialized these covariate-specific parameters randomly for fine-tuning. Because we used a version of the survey datasets that encoded each individual's location as a geographic region rather than as a state, we converted each state in the resumes data to be in one of four regions for pretraining: northeast, northcentral, south, or west. We also added a fifth "other" region for Puerto Rico and for when a state was missing in the original dataset. We also converted educational degrees to levels of experience: we converted associate's degree to represent some college experience and bachelor's degree to represent four-year college experience; we combined masters and doctorate to represent a single "graduate degree" category; and we left the other categories as they were.

**NLSY79.** The National Longitudinal Survey of Youth 1979 (NLSY79) is a survey following individuals born in the United States between 1957-1964. The survey included individuals who were between 14 and 22 years old when they began collecting data in 1979; they interviewed individuals annually until 1994, and biennially thereafter.

Each individual in the survey is associated with an ID, allowing us to track their careers over time. We converted occupations, which were initially encoded as OCC codes, into "occ1990dd" codes using a crosswalk (Autor & Dorn, 2013). We use a version of the survey that has entries up to 2014. Unlike the resumes dataset, NLSY79 includes three states corresponding to individuals who are not currently employed: unemployed, out-of-labor-force, and in-school. We include special tokens for these states in our sequences. We drop examples with missing occupation states. We also drop sequences for which the individual is out of the labor force for their whole careers.

We use the following covariates: years, educational experience, location, race/ethnicity, and gender. We drop individuals with less than 9 years of education experience. We convert years of educational experience into discrete categories: no high school degree, high school degree, some college, college, and graduate degree. We convert geographic location to one of four regions: northeast, northcentral, south, and west. We treat location as a static variable, using each individual's first location. We use the following race/ethnicities: white, African American, Asian, Latino, Native American, and other. We treat year and education as dynamic covariates whose values can change over time, and we consider the other covariates as static. This preprocessing leaves us with a dataset consisting of 12,270 individuals and 239,545 total observations.

**NLSY97.** The National Longitudinal Survey of Youth 1997 (NLSY97) is a survey following individuals who were between 12 and 17 when the survey began in 1997. Individuals were interviewed annually until 2011, and biennially thereafter.

Our preprocessing of this dataset is similar to that of NLSY79. We convert occupations from OCC codes into "occ1990dd" codes. We use a version of the survey that follows individuals up to 2019. We include tokens for unemployed, out-of-labor-force, and in-school occupational states. We only consider individuals who are over 18 and drop military-related occupations. We use the same covariates as NLSY79. We use

the following race/ethnicities: white, African-aAmerican, Latino, and other/unknown. We convert years of educational experience into discrete categories: no high school degree, high school degree, some college degree, college degree, graduate degree, and a special token when the education status isn't known. We use the same regions as NLSY79. We drop sequences for which the individual is out of the labor force for their whole careers. This preprocessing leaves us with a dataset consisting of 8,770 individuals and 114,141 total observations.

**PSID.** The Panel Study of Income Dynamics (PSID) is a longitudinal panel survey following a sample of American families. It was collected annually between 1968 and 1997, and biennially afterwards.

The dataset tracks families over time, but it only includes occupation information for the household head and their spouse, so we only include these observations. Occupations are encoded with OCC codes, which we convert to "occ1990dd" using a crosswalk (Autor & Dorn, 2013). Like the NLSY surveys, PSID also includes three states corresponding to individuals who are not currently employed: unemployed, out-of-labor-force, and in-school. We include special tokens for these states in our sequences. We drop other examples with missing or invalid occupation codes. We also drop sequences for which the individual is out of the labor force for their whole careers.

We consider five covariates: year, education, location, gender, and race. We include observations for individuals who were added to the dataset after 1995 and include observations up to 2019. We exclude observations for individuals with less than 9 years of education experience. We convert years of education to discrete states: no high school, high school diploma, some college, college, and graduate degree. We convert geographic location to one of four regions: northeast, northcentral, south, and west. We treat location as a static variable, using each individual's first location. We use the following races: white, Black, and other. We treat year and education as dynamic covariates whose values can change over time, and we consider the other covariates as static. This preprocessing leaves us with a dataset consisting of 12,338 individuals and 62,665 total observations.

## G   Experimental Details

**Model Specifications.**   We consider a first-order Markov model and a second-order Markov model (both without covariates) as baselines. These models are estimated by averaging observed transition counts. We smooth the first-order Markov model by taking a weighted average between the empirical transitions in the training set and the empirical distribution of individual jobs. We perform this smoothing to account for the fact that some feasible transitions may never occur in the training set due to the high-dimensionality of feasible transitions. We assign 0.99 weight to the empirical distributions of transitions and 0.01 to the empirical distribution of individual jobs. We smooth the second-order model by assigning 0.5 weight to the empirical second-order transitions and 0.5 weight to the smoothed first-order Markov model.

When we add covariates to the Markov linear baseline, we also include manually constructed features about history to improve its performance. In total, we include the following categorical variables: the most recent job, the prior job, the year, a dummy indicating whether there has been more than one year since the most recent observed job, the education status, a dummy indicating whether the education status has changed, and state (for the experiments on NLSY79 and PSID, we also include an individual's gender and race/ethnicity). We also add additive effects for the following continuous variables: the number of years an individual has been in the current job and the total number of years for which an individual has been in the dataset. In addition, we include an intercept term.

For the bag-of-jobs model, we vary the representation dimension $D$ between 256-2048, and find that the predictive performance is not sensitive to the representation dimension, so we use $D = 1024$ for all experiments.

We also compare to NEMO (Li et al., 2017), an LSTM-based method developed for modeling job sequences in resumes. We adapted NEMO to model survey data. In its original setting, NEMO took as input static covariates (such as individual skill) and used these to predict both an individual's next job title and their company. Survey datasets differ from this original setting in a few ways: covariates are time-varying, important covariates for predicting jobs on resumes (like skill) are missing, and an individual's company

name is unavailable. Therefore, we made several modifications to NEMO. We incorporated the available covariates from survey datasets by embedding them and adding them to the job embeddings passed into the LSTM, similar to the method CAREER uses to incorporate covariates. We removed the company-prediction objective, and instead only used the model to predict an individual's job in the next timestep. We considered two sizes of NEMO: an architecture using the same number of parameters as CAREER, and the smaller architecture proposed in the original paper. We found the smaller architecture performed better on the survey datasets, so we used this for the experiments. This model contains 2 decoder layers and a hidden dimension of 200.

We compare to two additional baselines developed in the data mining literature: job representation learning (Dave et al., 2018) and Job2Vec (Zhang et al., 2020). These methods require resume-specific features such as skills and textual descriptions of jobs and employers, which are not available for the economic longitudinal survey datasets we model. Thus, we adapt these baselines to be suitable for modeling economic survey data. Job representation learning (Dave et al., 2018) is based on developing two graphs, one for job transitions and one for skill transitions. Since worker skills are not available for longitudinal survey data, we adapt the model to only use job transitions by only including the terms in the objective that depend on job transitions. We make a few additional modifications, which we found to improve the performance of this model on our data. Rather than sampling 3-tuples from the directed graph of job transitions, we include all 2-tuple job transitions present in the data, identical to the other models we consider. Additionally, rather than using the contrastive objective in Equation 4 of Dave et al. (2018), we optimize the log-likelihood directly — this is more computationally intensive but leads to better results. Finally, we include survey-specific covariates (e.g. education, demographics, etc.) by adding them to $\mathbf{w}_x$, embedding the covariate of each most recent job to the same space as $\mathbf{w}_x$. We make similar modifications to Job2Vec (Zhang et al., 2020). Job2Vec requires job titles and descriptions of job keywords, which are unavailable for economic longitudinal survey datasets. Instead, we modify Equation 1 in Zhang et al. (2020) to model occupation codes rather than titles or keywords and optimize this log-likelihood as our objective. We also incorporate survey-specific covariates by embedding each covariate to the same space as $\mathbf{e}_i$ and adding it to $\mathbf{e}_i$ before computing Equation 2 from Zhang et al. (2020), which we also found to improve performance. We follow Dave et al. (2018) and use 50 embedding dimensions for each model, and optimize with Adam using a maximum learning rate of 0.005, following the minibatch and warmup strategy described below.

For the transformer models, we used model specifications similar to the generative pretrained transformer (GPT) architecture (Radford et al., 2018). This model includes a few extra modifications to improve training: we use 0.1 dropout (Srivastava et al., 2014) for the feedforward neural network weights, 0.1 dropout for the attention weights, and the GELU nonlinearity (Hendrycks & Gimpel, 2016) for all feedforward neural networks. Additionally, the model uses layer normalization (Ba et al., 2016) before the updates in Equation 13, after the update in Equation 16, and after the final layer's neural network update in Equation 17. We also considered other strategies for incorporating covariates into the model, including introducing them later in the transformer architecture and including different final-layer weights for each covariate. We found the way we incorporated covariates described in the paper resulted in the best predictions. We performed ablations separately for each model (since the non-transferred model is trained on fewer data points and did not require as many parameters). For the non-pretrained CAREER, we considered 4 and 12 layers, 64 and 192 embedding dimensions, 256 and 768 hidden units for the feedforward neural networks, and 2 or 3 attention heads (using 2 heads for $D = 64$ and 3 heads for $D = 192$ so that $D$ was divisible by the number of heads). We tried each combination of these parameters on NLSY79, and found that the model with the best validation performance had 4 layers, $D = 64$ embedding dimensions, 256 hidden units, and 2 attention heads. We used this architecture for the non-pretrained version of CAREER on all survey datasets. We performed a similar ablation for the pretrained version of CAREER, and found that 12 layers, 192 embedding dimensions, 3 attention heads, and 768 hidden units performed best. In total, this resulted in 5.6 million parameters for the full CAREER model.

We also considered a CAREER model that is pretrained on resumes but not fine-tuned. However, we found this model to be very poor at modeling occupations on survey data, because some occupations in the survey data (such as "unemployed") are never seen in the resume dataset. Thus, the model that is not fine-tuned assigns close to zero probability for these occupations. This underscores the importance of fine-tuning.

**Training.** We randomly divide the resumes dataset into a training set of 23.6 million sequences, and a validation and test set of 23 thousand sequences each. We randomly divide the survey datasets into 70/10/20 train/test/validation splits.

The first- and second-order Markov models without covariates are estimated from empirical transitions counts. We optimize all other models with stochastic gradient descent with minibatches. In total, we use 16,000 total tokens per minibatch, varying the batch size depending on the largest sequence length in the batch. We use the Adam learning rate scheduler (Kingma & Ba, 2015). All experiments on the resumes data warm up the learning rate from $10^{-7}$ to 0.0005 over 4,000 steps, after which the inverse square root schedule is used (Vaswani et al., 2017). For the survey datasets, we also used the inverse square root scheduler, but experimented with various learning rates and warmup updates, using the one we found to work best for each model. For CAREER with pretrained representations, we used a learning rate of 0.0001 and 500 warmup updates; for CAREER without pretraining, we used a learning rate of 0.0005 and 500 warmup updates; for the bag of jobs model, we used a learning rate of 0.0005 and 5,000 warmup updates; for the regression model, we used a learning rate of 0.0005 and 4,000 warmup updates. We use a learning rate of 0.005 for job representation learning and Job2Vec, with 5,000 warmup updates. All models besides were also trained with 0.01 weight decay. All models were trained using Fairseq (Ott et al., 2019).

When training on resumes, we trained for 85,000 steps, using the checkpoint with the best validation performance. When fine-tuning on the survey datasets, we trained all models until they overfit to the validation set, again using the checkpoint with the best validation performance. We used half precision for training all models, with the exception of the following models (which were only stable with full precision): the bag of jobs model with covariates on the resumes data, and the regression models for all survey dataset experiments.

The tables in Section 4 report results averaged over multiple random seeds. For the results in Figure 2a, the randomness includes parameter initialization and minibatch ordering. For CAREER, we use the same pretrained model for all settings. For the forecasting results in Table 2 and Table 1, the randomness is with respect to the Monte-Carlo sampling used to sample multi-year trajectories for individuals. For the wage prediction experiment in Table 3, the randomness is with respect to the sampling distribution of the test set.

**Forecasting.** For the forecasting experiments, occupations that took place after a certain year are dropped from the train and validation sets. When we forecast on the resumes dataset, we use the same train/test/-validation split but drop examples that took place after 2014. When we pretrain CAREER on the resumes dataset to make forecasts for PSID and NLSY97, we use a cutoff year of 2014 as well. We incorporate two-stage prediction into the baseline models because we find that this improves their predictions.

Although we do not include any examples after the cutoff during training, all models require estimating year-specific terms. We use the fitted values from the last observed year to estimate these terms. For example, CAREER requires embedding each year. When the cutoff year is 2014, there do not exist embeddings for years after 2014, so we substitute the 2014 embedding.

We report forecasting results on a split of the dataset containing examples before and after the cutoff year. To make predictions for an individual, we condition on all observations before the cutoff year, and sample 1,000 trajectories through the last forecasting year. We never condition on any occupations after the cutoff year, although we include updated values of dynamic covariates like education. For forecasting on the resumes dataset, we set the cutoff for 2014 and forecast occupations for 2015, 2016, and 2017. We restrict our test set to individuals in the original test set whose first observed occupation was before 2015 and who were observed to have worked until 2017. PSID and NLSY97 are biennial, so we forecast for 2015, 2017, and 2019. We only make forecasts for individuals who have observations before the cutoff year and through the last year of forecasting, resulting in a total of 16,430 observations for PSID and 18,743 for NLSY97.

**Rationalization.** The example in Figure 3 shows an example of CAREER's rationale on PSID. To simplify the example, this is the rationale for a model trained on no covariates except year. In order to conceal individual behavior patterns, the example in Figure 3 is a slightly altered version of a real sequence. For this example, the transformer used for CAREER follows the architecture described in Radford et al. (2018). We find the rationale using the greedy rationalization method described in Vafa et al. (2021). Greedy

rationalization requires fine-tuning the model for compatibility; we do this by fine-tuning with "job dropout", where with 50% probability, we drop out a uniformly random amount of observations in the history. When making predictions, the model has to implicitly marginalize over the missing observations. (We pretrain on the resumes dataset without any word dropout). We find that training converges quickly when fine-tuning with word dropout, and the model's performance when conditioning on the full history is similar.

Greedy rationalization typically adds observations to a history one at a time in the order that will maximize the model's likelihood of its top prediction. For occupations, the model's top prediction is almost always identical to the previous year's occupation, so we modify greedy rationalization to add the occupation that will maximize the likelihood of its *second-largest* prediction. This can be interpreted as being equivalent to greedy rationalization, albeit conditioning on switching occupations. Thus, the greedy rationalization procedure stops when the model's second-largest prediction from the target rationale is equivalent to the model's second-largest prediction when conditioning on the full history.

