# OpenReview forum: "CAREER: A Foundation Model for Labor Sequence Data"
_TMLR — Accepted by TMLR_

### Review · Reviewer_7qrm · 2023-06-19

**Summary Of Contributions:**

The work considers an important problem on the prediction of future occupation of a person using a large database of CVs and longitudinal surveys. They adopt the transfer learning approach to solve the problem at hand, as the CV's database is much large, but less accurate.

**Audience:**

Yes

**Broader Impact Concerns:**

The possible concerns are described by authors in the paper. The description there is good. I have nothing to add.

**Claims And Evidence:**

No

**Requested Changes:**

Right now the paper is better suited for a journal dedicated to analysis of labour market or some applied econometrics journal. To make it work for the audience of machine learning specialist I suggest the following changes.
* The authors should specify all hyperparameters they use, the detailed model architecture, etc. The results should be fully reproducible.
* Ablation studies would of separate interest: how we can improve the model by using a different language model, a different transfer learning strategy, etc.


**Strengths And Weaknesses:**

Strengths:
* nice problem to be solved
* superior quality to a single-source models
Weaknesses:
* No open source data released
* Lack of implementation details: how exactly the transfer was performed? how exactly were features from the resumes extracted?
* No ablation study

---

> ### Author Response · Authors · 2023-10-06
> **Author response**
>
> Thank you for your evaluation and comments. We are glad you found the paper to be addressing an important problem and well-written, and that the experimental results were convincing.
>
> **Hyperparameters/reproducibility:** The appendix of our paper includes details about the hyperparameters, the detailed modeling architecture, details for the transfer learning, and how features are extracted from resumes. Moreover, we've included code in the supplementary material so our results are reproducible. We've linked to the appendix earlier in the paper in the latest revision. If there are particular questions not addressed in this appendix, could you kindly specify? We're committed to ensuring clarity and consistency throughout the paper.
>
> **Ablation:** Our appendix also has information about ablation studies involving different model architectures. The main results in the paper include a comparison of 8 different models, including two different classes of language models (NEMO/LSTM vs CAREER/transformer) and three ablations of CAREER. Taken together, we believe these ablations provide a detailed comparison of possible models.

---

### Review · Reviewer_8Vvr · 2023-07-24

**Summary Of Contributions:**

The paper pretrains a transformer model on a large dataset of resumes to predict sequences of job changes. The pretrained model is then fine-tuned on 3 smaller economic datasets. The finetuned transformer model outperforms existing baselines.

**Audience:**

Yes

**Claims And Evidence:**

Yes

**Requested Changes:**

Did the authors consider any other ways to incorporate covariates, e.g. as separate embeddings?

Did the authors test models with fewer parameters? Based on the corpus size and complexity of the problem, 12 layers may be overkill.

The paper mentions the two stage model as being important noting that making previous approaches two-stage improves there performance, but it’s not clear to my that this would necessarily be the case with a transformer model. Did the authors test a version of the approach that is one-stage but includes pertaining?

**Strengths And Weaknesses:**

Strengths
- The paper is a well written empirical study of using transformer models and the pretrain / fine-tune paradigm for job change prediction
- The resulting model outperforms existing approaches
- The appendix includes extensive training details

Weaknesses
- The paper doesn't include technical ML innovations; it relies on off the shelf methods. The paper might be more appropriate in a more applied venue than TMLR.
- There could be more experiments considering different modeling decisions.
- The model size might be overkill.

---

> ### Author Response · Authors · 2023-10-06
> **Author response**
>
> Thank you for your evaluation and comments. We are glad you found the paper to be well-written and the experimental results to be convincing.
>
> **Other ways of incorporating covariates:** Yes, we experimented with other methods for incorporating covariates, including introducing them later in the transformer architecture and including different final-layer weights for each covariate. We found the way we incorporated covariates to result in the best predictions. We have added more details to the paper.
>
> **Fewer parameters:** Yes, as described in the appendix, we considered a range of models that ablated model size and complexity. We found the 12-layer model to result in the best predictions.
>
> **Two-stage:** Good question. We found that two-stage prediction was more important for the non-pretrained transformer model than for the pretrained one. Still, as described in the paper, two-stage prediction improves all baselines, sometimes significantly.
>
> **Applications:** Your review states that our paper "might be more appropriate in a more applied venue than TMLR". First, as described in [TMLR's submission guidelines](https://jmlr.org/tmlr/editorial-policies.html), TMLR invites papers that contain "accounts of applications of existing techniques that shed light on the strengths and weaknesses of the methods." Moreover, we believe the novelty of our work is in developing a transfer learning strategy in an econometric setting where it has not been considered. In addition to the modifications to the transformer architecture described in the paper, this approach requires identifying a source of pretraining data (passively-collected resumes), which, on its face, is quite different from economic surveys, yet results in state-of-the-art predictions. This enables transfer learning to be used in important econometric tasks where it has not been considered, such as estimating gender wage gaps, racial disparities in unemployment, and measures of occupational mobility. As Reviewer bDnb stated: the method "[opens] up pathways for labour economists to extract information from large resume databases and use them in generic forecasting tasks."

---

### Review · Reviewer_bDhb · 2023-09-26

**Summary Of Contributions:**

The authors propose a new transformer-based for representation learning of job sequence data (CAREER), which can then be ported to answer questions in labour economics via transfer learning (i.e. wage forecasting). This is achieved in a two step process. First, CAREER is trained to forecasts job sequences from large databases of online resume data —  demonstrating better prediction performance compared to other comparable benchmarks. Then, feature vectors from CAREER are combined with other observed covariates in a linear prediction model, and are shown to improve wage forecasts over just covariates alone.

**Audience:**

Yes

**Claims And Evidence:**

Yes

**Requested Changes:**

1 & 2 in the weaknesses section would be critical for me, while 3 is more of a good to have.

**Strengths And Weaknesses:**

Strengths
---
The approach represents a very concrete application	of representation and transfer learning — opening up pathways for labour economists to extract information from large resume databases and use them in generic forecasting tasks. The paper is also clearly presented, both in terms of the models for career trajectories and wage forecasts, as well as the structure of the transformer model itself. The performance improvements observed both for job sequence forecasting and the wage regression validate the method proposed.


Weaknesses
---
1) While results are stronger on average, uncertainty estimates would be required for Table 1 & 2 to help evaluate the significance of the improvements.
2) For the transfer learning problem, it would also be good to evaluate the performance of the transformer model trained directly from the wage data — as a way to disentangle if gains are due to the model used or from the transfer learning approach.
3) For non-experts in economics, additional background on direct applications of wage forecasting would also help strengthen the case/importance of the model.

---

> ### Author Response · Authors · 2023-10-06
> **Author response**
>
> Thank you for your careful evaluation and comments. We appreciate your comments about the strengths of the paper's experimental results, clarity, and potential impacts.
>
> **Uncertainty estimates:** We have now added uncertainty estimates to all of the tables in the main text of the updated revision. These results reaffirm the strengths of CAREER's results.
>
> **Non-transferred baseline for wage prediction:** Good idea. We included the baseline you suggested. This additional baseline shows that while including a non-pretrained transformer improves predictions over the econometric baseline, pretraining further expands the advantage over all years. Here are the full results:
>
> |                                             | Overall            | 1990-1999          | 2000-2009          | 2010-2019          |
> |---------------------------------------------|--------------------|--------------------|--------------------|--------------------|
> | Full specification from Blau et al., 2017   | 0.431 ± 0.000      | 0.424 ± 0.000      | 0.392 ± 0.000      | 0.459 ± 0.000      |
> | Full specification + CAREER (not pretrained)| 0.493 ± 0.000      | 0.498 ± 0.000      | 0.453 ± 0.000      | 0.517 ± 0.000      |
> | Full specification + CAREER (pretrained)    | **0.504 ± 0.001**  | **0.504 ± 0.001**  | **0.461 ± 0.000**  | **0.532 ± 0.000**  |
>
>
> **Additional background on applications of wage prediction:** Wage prediction models underpin many economic estimates, such as the gender wage gap [1], wage effects in labor market discrimination [2], and union wage premiums [3]. Estimating all of these quantities involves building a model to predict an individual's wage. By making more accurate wage predictions, CAREER can result in more accurate estimates of these quantities. We've added more details in the paper to make this background more clear.
>
> [1] Blau and Kahn. The gender wage gap: Extends, trends, and explanations. 2017.
> [2] Neal and Johnson. The role of premarket factors in black-white wage differences. 1996.
> [3] Linneman, Wachter, and Carter. Evaluating the evidence on union employment and wages. 1990.

---

### Review · Reviewer_o9LQ · 2023-10-03

**Summary Of Contributions:**

This paper presents an approach for improving predictions for econometric models via pretraining using semi-structured data derived from a resume database with finetuning on longitudinal survey datasets. A transformer-based model architecture is adapted for categorical data modalities and is shown to improve performance over previous methods when predicting and forecasting occupations. In addition, the internal representations learned through the pretraining approach are shown to be beneficial for another downstream economic prediction task.

**Audience:**

Yes

**Broader Impact Concerns:**

I have none beyond those already addressed and discussed by the authors in the included broader impact statement and the expressed weaknesses I have recorded above.

**Claims And Evidence:**

No

**Requested Changes:**

**Major**

More detail about the datasets (especially the demographic features used as covariates in all experiments).

Additional prediction performance metrics such as AUROC/AUPRC or F1 score to evaluate the quality of the compared models.\

Additional NEMO baseline that is pretrained on resume database, then finetuned on the survey datasets.

Additional baseline removing covariates from CAREER.

Some analysis on the "strength" of the representations learned rather than some implied validity of their construction based on the complexity of the model.

A model ablation of CAREER without the covariates included.

Further analysis of the claims made about the necessity of the transformer architecture (using information over the whole sequence of career history).

**Minor**

More clarity in writing about the domain and the specifics of the kinds of predictive analyses being used with the data. This would be to better frame the contributions of the proposed model.

CAREER is clearly an acronym (as stated in footnote 2). It should be spelled out as early as the modeling name/approach is used in Section 1 to help frame the contributions and claims made at the end of the introduction.

Additional experimental analysis comparing CAREER's prediction of occupational change with prior approaches.

Restructuring Section 2 as outlined above to improve clarity.

A baseline of zero-shot prediction performance of the the pretrained CAREER model without finetuning.

**Strengths And Weaknesses:**

**Strengths**

The proposed algorithmic approach for developing informative representations of career history (via what is called CAREER) is well grounded in prior use of transformer models in a pre-training/fine-tuning paradigm. The adjustments made to incorporate categorical data sequences through initial embedding layers is interesting, although seems to be fairly limited by how expressive these embeddings can be as they are effectually a key-value store indexed by integer quantities.

I appreciate the delineation made by the authors between expansive resume databases and carefully curated longitudinal econometric occupational surveys. It was made apparent (while the connotation could be made more explicit and direct) that policy decisions and generalizable claims cannot be made from the use of these large resume databases as they are not representative of the population at large. What is missing from this very clear dichotomy is a good, clear and direct discussion of why econometric models need well curated survey datasets. I'll point to additional issues with the presentation of the data/problem formulation below.

The paper does a nice job outlining prior approaches and methodologies for occupation prediction within economics including some interesting recent approaches such as NEMO that are built to learn from unstructured resume datasets.

For the most part, the setup and description of how the proposed CAREER model is constructed is clearly understood. Perhaps one addition to Figure 1 would be to include annotations of the FFN between $\tilde{h}^{(1)}t$ and $h^{(2)}_t$ similar to how the attention weights are shown.

There is a variety of experiments to demonstrate the utility of using pretrained representations, adapted for the occupation prediction task. I was especially glad to see the experiment represented in Figure 2B that shows the effect on pretraining volume on final model prediction performance. The forecasting and wage regression experiments were also very interesting and good demonstrations of how the pretrained representations can be useful (despite some major questions that arose from how they were presented, details below).


**Weaknesses**

I fear that there are several relatively major weaknesses that lessen the excitement I have for this paper and the presented empirical results. Some are easily fixed (more thorough and precise definitions and writing) while others may take a fair amount of effort on the authors' behalf (additional baseline experiments and model ablations). As a summary, there are too few details about the motivating foundations of the work, the datasets used, the modeling approach, the experimental setup and the analyses presented. As a result I am not entirely sure that I can confidently state that the claims made are adequately supported by the paper. In the following I will try to highlight where I am concerned and how the authors may be able to revise the paper to help improve my estimation of the work. This being said, I feel that the insights provided by the paper are good, it can however be greatly improved through more precise language and thorough empirical validation. This work will be of interest to the community using ML within economics as well as the broader ML community for yet an additional domain where scaling pretraining enables improved predictive with transformer models.

_Motivating foundations_

I found the writing and descriptions about the econometric setting to be insufficient and vague. There are not enough details shared about the kinds of predictive tasks that are prevalent in this community and why they are important. This extends to any motivations about the datasets that are used and how analyses of these survey datasets support claims and potential policy decisions (if any are to be had). Greater clarity about the domain specific usage of the data and why economists take such great effort to develop survey datasets would go a long way to improve the apparent significance of the contributions made by this paper.

Many of the motivations behind the experiments are vague and seem to rest on a solitary example without additional information or detail about why the experiments are chosen and how informative they may be in view of the intended use case.

_Datasets_

It's not clear how the datasets are formed, what the size of J or $N_c$ (the number and scale of the covariates), etc. are. The discussion or information provided about the datasets (both the resume and longitudinal survey data) is surprising. Without these details it is unclear what is actually being modeled or the overall scale of the problem. For example, how do the data distributions of job categories or types compare between the pretraining data and each of the survey datasets? Is J consistent across the whole paper? If not, what kinds of support can be found in the pretraining data that overlaps with what is represented in the survey data? This isn't a critical omission as the methods used to construct CAREER and analyze are the clear focus of the paper, but I would rate the omission as a major issue that should be corrected to improve the presentation of the paper.

In connection to the concerns raised previously, there is no indication of how long the career sequences are in each dataset on average (e.g. what is the distribution of T?). I know that the perplexity metric tries to average over this component but it could be a major factor in the forecasting performance as errors will compound for longer career durations. (This is a shared concern with the analyses provided in the paper) If it's true that some of the datasets have longer sequences than others, that could explain why the finetuning task takes longer on datasets with equivalent numbers of individuals.

It is claimed at the end of Section 3 that this paper "identif[ies] a novel pretraining corpus". This isn't necessarily true, is it? It is stated that "NEMO, an LSTM-based model that is trained on large resume datasets". While NEMO may not be immediately applied to the pretraining/finetuning paradigm, it may not be wholly accurate to state that the use of the same (or similar) dataset for pretraining is novel.

_Modeling approach_

It is very unclear how the predicted job change indicator $s_t$ is used to inform the input $y_{t-1}$ at the next timestep as indicated in Figure 1.

There's an asymmetry in how the covariates are introduced and described and how they are ultimately used in the experiments. In fact, it's unclear between experiments what covariates are used and if they are at all. This is not insignificant. _This inconsistency is a problem that should be addressed by the authors if this paper is to be considered for publication_.

Are there any additional regularizations used to keep the representations $h_t$ from collapsing to trivial modes? With the amount of data and number of model parameters, there is some risk for model collapse. This highlights one of my complaints of the presentation of this paper. Several times the representations are claimed to be "complex" or "strong" but there is no precision or justification of those terms. It appears that this nomenclature is used implicitly from the complexity of the model architecture. There is no analysis of the representations and what is being encoded within them. The only indication that we have that anything is being learned is model performance. There are no clear indicators that the representations differentiate between different career paths or how the effect of individual covariates have on the representation. Without any analysis or justification of how the representations are formed and what they encode, I cannot place any weight on the language or claims made about them. It's highly possible that pretrained representations using NEMO could be equally as performant on the dataset not to mention more rudimentary clustering approaches. This significantly impacts how I view the empirical support of the claims made by the paper.

_Experimental Setup_

It would be very useful to recast the apparent strength of the transformers learned representations $h_t$ in the context of additional predictive tasks (beyond job prediction) to better compare with prior approaches outlined in Section 3. With the two-stage regression technique of CAREER, occupational change is already modeled. It would be instructive to see how that sub-task performs relative to the the cited approaches of Kambourov & Monovskii (2008) and Guvenen et, al. (2020).

As mentioned above, there are concerns about whether the apparent gains seen with CAREER can be contributed mostly to the transformer architecture more than the pretraining approach. There are no empirical comparisons with an alternative architecture that is provided the same training / evaluation procedure (two-stage regression from the embedded representation of career path). The suggested NEMO pretraining baseline could be a way to validate the use of a more complex transformer architecture.

An additional baseline that would help reinforce the overall empirical approach used in the paper would be to pretrain CAREER but not finetune. To demonstrate the zero-shot capabilities of the model based on the resume data alone. This could open up a really interesting analysis of the questions raised above about the representative support of the pretraining dataset (e.g. are there certain individuals that are included in the survey data for which not finetuning does well on? Are there inequalities that arise without the finetuning step? Do those inequalities persist after finetuning?)

It would really help if the objective or loss function was formally included in the main paper. This would help round out Section 2.3.

I found the ordering of Section 2 to be a little confusing. The most important formulations of the CAREER model are presented in Section 2.2 where there is not much context behind how they would be used. Then after so much information is given in Section 2.3 about architecture and training approach, there is a subtle reference to the previous subsection. I think that the Section 2 would be improved if the order of Section 2.2 and 2.3 were swapped. I also think that the "Transfer Learning" subsubsection could be elevated into a new Section 2.4 to help emphasize the contributions and claims made in the paper.

The specific tasks "career prediction" and "forecasting" are not adequately defined at the outset of Section 4. This is another example where just a little more detail would go a long way. This issue occurs again when the downstream auxiliary tasks (wage prediction in this paper) are introduced. There is reference to multiple types of economic analyses that could be run but there is no mention of what these are. I found a lot of the language around these type of context-setting portions to be frustratingly vague.

_Experimental Analysis_

While a helpful comparative metric, perplexity is difficult to gain any insight from. I believe that including additional metrics such as AUROC/ AUPRC or F1 score would be far more informative of the quality of the predictions being made by the compared models. This is particularly true when considering the forecasting task presented in Table 1. I have no way to assess how reliable the models are as perplexity gives me no indication of how immediately accurate the predictions are.

It's not sufficient to only report the mean performance of the prediction metrics in Tables 1 and 2 without confidence intervals as is done in the table contained in Figure 2A. Without these the significance of the performance improvement seen with CAREER cannot be assessed. Their exclusion seems somewhat intentional?

While NEMO is included as a baseline, it's unclear whether it was pretrained using the resume dataset as it was developed for. So, it's not entirely clear whether its current use is a fair comparison of the model's performance. It would be instructive to see if NEMO (and other baseline modeling approaches) would have better performance than their vanilla counterparts if they were additionally exposed to more data in a pretraining setting and how that would compare after finetuning to the variants of CAREER presented in Figure 2A. Without this baseline, it's unclear whether the performance gains are drawn from the architecture (e.g. model complexity/expressivity) or the training paradigm. This is perhaps the primary major concern of mine regarding the evaluation supporting the claims made in the paper.

Given the amount of discussion around the assumed benefit of using a more complex model that breaks down prior work's Markov assumption, it is surprising that the authors did not analyze the attention weights to demonstrate any evidence of their claims on this premise. It would be really interesting to see how far the prediction of the current time step relies on previous employment _and_ the sensitivity of the prediction on the included covariates. Without any analysis along these lines, I cannot trust the claims made about the necessity of non-Markovian modeling. I cannot just take the claim at face value because of the prediction metrics are "better". We can expect there to be more rigor in the experiments to validate the claims made.

---

> ### Author Response · Authors · 2023-10-06
> **Author response (part 1)**
>
> Thank you for your thorough evaluation and insightful comments. We are glad you found the paper to be well-written and the experimental results to be interesting.
>
> **Motivating foundation:** We appreciate your point about emphasizing the motivation. Econometric analyses often rely on representative small survey datasets, which are crucial for forming analyses that generalize to larger populations. These analyses frequently involve building predictive models involving sequences of occupations. For example, models that predict an individual's next occupation are used to estimate racial differences in unemployment [1] and changes in occupational mobility [2]. Models that predict wage using features from an individual's career are used to estimate quantities such as the gender wage gap [4], wage effects in labor market discrimination [5], and union wage premiums [6]; see the end of Section 4 for discussion about how our approach can be incorporated into gender wage gap estimation. In general, if more accurate predictive models can be developed from small survey datasets, the above quantities can be estimated more reliably. The exact details for how these models are incorporated into econometric analyses is beyond the scope of this paper but they are described in detail in the citations. We'll make this more clear in the paper, and we will also incorporate your additional suggestions about writing (introducing the CAREER acronym earlier on, restructuring section 2, explicitly writing the loss function).
>
> **More detail about the datasets:** The beginning of section 4 along with Appendices E and H include detailed information about how the datasets are constructed and the covariates included. These sections include information about the size of J (330 total occupations), the number of covariates included (5 for each survey dataset) and how the covariates are constructed. The demographic covariate refers to an individual's ethnicity/race, which is crucial for economic analyses involving racial disparities, like estimating racial wage gaps or racial differences in unemployment. Tables 7 and 8 in the appendix include information about differences between pretraining data and the survey datasets. We'll move this information earlier in the paper to emphasize them.
>
> **Covariates:** Your review states "It's unclear between experiments what covariates are used and if they are at all. This is not insignificant. This inconsistency is a problem that should be addressed by the authors if this paper is to be considered for publication." Appendix G provides a comprehensive breakdown of which covariates correspond to each dataset. If there are particular questions not addressed in this appendix, could you kindly specify? We're committed to ensuring clarity and consistency throughout the paper.
>
> **Prediction performance metrics:** Our primary evaluation metric is perplexity, aligning with standard practice for likelihood-based sequence models in the transformer literature. Due to the categorical nature of our predictions, AUC and F1 score aren't directly applicable. Nevertheless, we performed an experiment that evaluated each model's ability of making the binary prediction of whether an individual stays in their current occupation or transitions to another one. From these binary predictions, we computed AUC, where CAREER has a clear advantage. We've added this table to the paper:
>
> | Model                   | PSID            | NLSY79          | NLSY97          |
> |-------------------------|-----------------|-----------------|-----------------|
> | Markov regression       | 0.701 ± 0.001   | 0.732 ± 0.000   | 0.680 ± 0.001   |
> | NEMO                    | 0.706 ± 0.004   | 0.731 ± 0.002   | 0.664 ± 0.001   |
> | Job rep. learning       | 0.679 ± 0.003   | 0.704 ± 0.000   | 0.651 ± 0.001   |
> | Job2Vec                 | 0.681 ± 0.005   | 0.705 ± 0.000   | 0.651 ± 0.001   |
> | Bag-of-jobs             | 0.690 ± 0.002   | 0.706 ± 0.000   | 0.661 ± 0.001   |
> | CAREER                  | **0.741 ± 0.001**   | **0.761 ± 0.000**   | **0.691 ± 0.001**   |

---

> > ### Author Response · Authors · 2023-10-06
> > **Author response (part 2)**
> >
> > **Pretrained NEMO baseline:** We opted for a non-pretrained NEMO as a baseline primarily due to the central contribution of this paper: the application of transfer learning to model small survey datasets. Consequently, a "fair comparison" isn't against a transferred version of NEMO but against its conventional, non-transfer learning approach. Your review mentioned concerns about the source of performance gains – whether they stem from the architecture itself (like model complexity/expressivity) or the training paradigm. To clarify this, Figure 2a demonstrates the significant role played by both factors, with Figure 2b offering a more granular view via the scaling law.
> >
> > Incorporating your feedback, we also tested NEMO with transfer learning, with pretraining details identical to CAREER's. While pretraining does improve NEMO's performance, CAREER continues to outperform. This reaffirms the importance of transfer learning:
> >
> > | Model                   | NLSY79          | NLSY97          | PSID            |
> > |-------------------------|-----------------|-----------------|-----------------|
> > | NEMO (not pretrained)   | 12.82 ± 0.04    | 18.38 ± 0.08    | 17.58 ± 0.04    |
> > | NEMO (pretrained)       | 11.72 ± 0.02    | 14.70 ± 0.01    | 14.25 ± 0.02    |
> > | CAREER                  | **11.32 ± 0.00**    | **14.15 ± 0.03**   | **13.88 ± 0.01**    |
> >
> >
> > **Baseline with covariates:** This is a good suggestion to consider baselines with varying levels of covariates. We trained models that do and do not contain each covariate. The most important covariate for predictive advantage is year, followed by education, and then the demographic/gender/location variables. We will include this information in the updated paper.
> >
> > **Strength of representations:** We evaluate the complexity of the learned representations using predictive approaches. The paper provides evidence that the complex model makes better predictions than the simpler baselines, both for the objective it is trained for (modeling occupations) and for an objective it isn't trained for (modeling wages). We also provide qualitative examples of the representation quality in Figures 3 and 5 in the appendix. All these results underscore the significance of our complex model within this research context. We believe that more in-depth analyses about the learned representations, such as those based on probing experiments, fall beyond the scope of this paper.

---

> > > ### Comment · Reviewer_o9LQ · 2023-10-08
> > > **comparison with pretrained NEMO**
> > >
> > > I think there may be some confusion over the notion of "fair" comparison vs. what the original method provides. Perhaps, that is due to a fault in my writing. My concern rests almost entirely that the proposed CAREER is given access to significantly more data during the pretraining phase which may be the cause of the major advantages seen in the empirical results rather than the specific architectural choice of the transformer model.
> > >
> > > The results shared in the author response do help designate that pretraining NEMO helps significantly! It's also really neat to see that CAREER maintains the advantages it does over pre-trained NEMO, which helps to confirm that the model and algorithmic choices underlying CAREER are important in this econometric task. Thus, the contributions of CAREER are more than just that transfer learning is important, but that the non-markovianity of the transformer architecture appears to also be important. The concerns/questions I raised about this second aspect were to suggest additional ways that one could analyze the degree to which the non-markovianity of CAREER contributes to the improvements seen in the included experimental results, I address the author's response to this point below.

---

> > ### Author Response · Authors · 2023-10-06
> > **Author response (part 3)**
> >
> > **Studying attention weights/necessity of non-Markov:** The NLP domain has largely moved away from relying on attention-based explanations, primarily because they often lack fidelity [6, 7, 8] and their handling of token information can obfuscate clear interpretations [9, 10]. Consequently, we have chosen to emphasize predictive performance as the primary evidence for the superiority of non-Markovian models. As our findings indicate, when compared to Markov models, CAREER improves the sequence modeling perplexity by 25-30% across all datasets (as illustrated in Figure 2a). This improvement is also mirrored in predictions spanning different transition types (refer to Tables 5 and 6) and in wage predictions, which show a 15% enhancement over existing baselines (Table 3). Figure 2a includes a series of baselines that vary the linearity and Markov assumptions (job rep. learning is Markov but learned with embeddings; bag-of-jobs is both non-Markov and learned with embeddings). These collective insights underscore the critical role of non-Markovian modeling in our context.
> >
> > **"Novel pretraining corpus":** While the data mining community has modeled resume datasets directly with ML methods, we are the first to use these datasets in a transfer learning approach to target *survey data*. This survey data is crucial for estimating economic quantities; see the examples in the related work. We'll clarify this in the next version of the paper.
> >
> > **Job change indicator s:** Equations 1-3 describe the data generating process as a probabilistic model. As discussed in the text, this variable is marginalized during training. If there are specific questions about this variable, could you provide further information? We aim for comprehensive clarity in our paper.
> >
> > **Uncertainty estimates:** The omission of standard errors was primarily due to space constraints in the original format. To address this, we've revised the presentation by splitting the tables into three (now labeled as Tables 1-3) to make room for these metrics. For all tables, the uncertainty estimates are small and reinforce CAREER's predictive advantage over baselines.
> >
> > **Zero-shot prediction of pretrained CAREER without finetuning:** This is a good suggestion. CAREER without fine-tuning performs very poorly at modeling occupations on survey datasets because some occupations in the survey data (such as "unemployed") are never seen in the resume dataset used for pretraining. Thus the perplexity is about 10x worse than any model, speaking to the necessity of fine-tuning. We'll add this to the paper.
> >
> >
> > [1] Hall. Turnover in the labor force. 1972.
> > [2] Kambourov and Manovskii. Rising occupational and industry mobility in the United States: 1968-97. 2008.
> > [3] Blau and Kahn. The gender wage gap: Extends, trends, and explanations. 2017.
> > [4] Neal and Johnson. The role of premarket factors in black-white wage differences. 1996.
> > [5] Linneman, Wachter, and Carter. Evaluating the evidence on union employment and wages. 1990.
> > [6] Jain and Wallace. Attention is not explanation. 2019.
> > [7] Serrano and Smith. Is attention interpretable? 2019.
> > [8] Bastings and Filippova. The elephant in the interpretability room: Why use attention as explanation when we have saliency methods? 2020.
> > [9] Brunner et al. On identifiability in transformers. 2019.
> > [10] Kobayashi et al. Attention is not only a weight: Analyzing transformers with vector norms. 2020.

---

> > > ### Comment · Reviewer_o9LQ · 2023-10-08
> > > **Continuing thoughts**
> > >
> > > I am more or less satisfied with the discussion here regarding non-markovianity being useful based on performance metrics. I was mostly hoping to suggest additional analyses that could be run to validate this through possible case studies / sensitivity analyses given changes in covariates, etc.
> > >
> > > My first concern/confusion about the job change indicator that is PREDICTED (right?) is based primarily on the arrow drawn from s to y in Figure 1. If this is more of an implicit dependence (meaning that there is/isn't a change in y based on the previous step's s indicator) then just adding a clarifying sentence in the corresponding section of text would be sufficient.

---

> > > > ### Author Response · Authors · 2023-10-08
> > > > **Response to new comments from Reviewer o9LQ**
> > > >
> > > > Thank you for your insightful comments and suggestions.
> > > >
> > > > **Covariates:** We understand the initial confusion surrounding the use of covariates. While the resume and survey datasets have different covariates available (e.g. resumes don't contain gender), the covariates are consistent across all survey datasets. As such, each model trained to predict occupations on each survey dataset uses the same covariates. We apologize that this wasn't clear in the original paper, and we will clarify it so it is more clear. We will provide clearer explanations in the revised paper.
> > > >
> > > > **Prediction metrics:** While many use-cases of occupation models concern predicting next occupations, others, such as modeling stepping-stone occupations [1] or using forecasted long-term trajectories (as in Tables 1-2) as surrogate indices [2], require modeling long-term career trajectories (analogous to sequence generation in NLP). This is why we focus on perplexity, which is also a useful metric for predicting next occupations since it is essentially a geometric average of next-occupation likelihoods. Still, we believe the AUC metrics you suggested (and whose results we included above) are a useful alternative metric and believe the paper is stronger with their inclusion. We will clarify all these points in the paper.
> > > >
> > > > **NEMO comparison:** We're glad you found the additional pretrained NEMO model to be insightful. We think the paper is stronger with this additional baseline. Thank you for suggesting it.
> > > >
> > > > **Non-Markovianity:** We are pleased that the provided performance metrics addressed your concerns regarding the utility of non-Markov baselines.
> > > >
> > > > **Job change indicator:** Yes, you are correct that the job change indicator is *predicted* from previous jobs. We will clarify this implicit dependence as per your suggestion.
> > > >
> > > > We truly appreciate your thorough engagement with our work. We think the paper has become stronger thanks to your suggestions.
> > > >
> > > > [1] Jovanovic and Nyarko. Stepping-stone mobility. 1997.
> > > > [2] Athey et al. The Surrogate Index: Combining Short-Term Proxies to Estimate Long-Term Treatment Effects more Rapidly and Precisely. 2019.

---

> > ### Comment · Reviewer_o9LQ · 2023-10-08
> > **Re: Covariates used in experiments and evaluation metrics**
> >
> > Hello! Thanks for your thorough response to my review!
> >
> > **Covariates**
> >
> > As a quick point re: my comment on how unclear the use of covariates are in the experiments. Based on my understanding of what was written in the paper, the number of covariates used in each analysis (using the same dataset) seemed to be inconsistent. I think being very clear at the outset of the experimental section about which covariates are used for each dataset (and phase of pre-training + finetuning) would go a long way toward addressing my concern. I understand if there are asymmetries between the resume data and the survey data which may constrain the covariates used in the fine-tuning step. But this should be more clearly stated in the main body of the paper.
> >
> > **Prediction metrics**
> >
> > My concerns about the evaluation metrics rest on the fact that the optimization of the transformer architecture is not for sequence generation (as is done in NLP) but for the prediction of the next occupation (let me know if that is not accurate). The fact that NLP uses perplexity is perhaps irrelevant in this regard? Now, if the proposed CAREER model was in use to generate full job histories, then I feel that perplexity would be an adequate measure of performance. There just seems to be an inconsistency here. This isn't an overly disqualifying concern but some clarity in the paper that offers full justification of the metrics used to compare methods really would go a long way.

---

### Author Response · Authors · 2023-10-06
**General author response**

We thank the reviewers for their careful evaluation and feedback. We are glad you found that our proposed method to have convincing experimental results and thought our paper "opens up pathways for labour economists to extract information from large resume databases and use them in generic forecasting tasks" (bDhb). Moreover, we're glad all reviewers found the paper to be well-written.

In response to the reviews, we've also updated the paper by:
- adding AUC metrics for predicting transitions
- including more years for the wage prediction experiment
- incorporating an additional non-pretrained baseline for wage prediction
- adding more uncertainty estimates
- clarifying experimental details in the main text

We have responded to each reviewer individually.

---

### Author Response · Authors · 2023-12-29
**Camera-ready version and detailed changes (1/2)**

Dear reviewers and editorial team,

Thank you for your insightful comments and discussion, which have greatly improved the paper. Our paper has been updated to incorporate the reviewer feedback. We are also including [code](https://github.com/keyonvafa/career-code) and a [short video](https://www.youtube.com/watch?v=BLgfbNeTLCI).

Below, we detail how the reviewer suggestions have been addressed:

_The motivation of the approach was considered insufficient and lacking detail about the data use in the econometric domain, the typical predictive tasks, their importance (including background on direct applications of wage forecasting [bDhb]), and how informative the experiments are for the use case [o9LQ]. Authors mentioned some discussion in a response._

We've modified Section 1 and Section 4 to make the motivation more clear. Section 1 now makes concrete the applications that benefit and quantities that can be more reliably estimated with more accurate occupation models. In Section 4, we've added a discussion about how wage prediction is useful for econometric analyses like estimating gender wage gaps, and the analysis is connected to prior work. We've also included more discussion about baselines, and changed the metric for wage prediction to R^2, so it is consistent with how the results have been reported in the econometrics literature.

_Lack of detail about data sets was criticized along a number of aspects: --- how they are formed, [o9LQ] --- numbers of occupations and of covariate values, [o9LQ] --- career sequence lengths, [o9LQ] --- distribution similarities/differences between pretraining data and survey data sets, [o9LQ] --- consistency of the number of occupations across data in the paper [o9LQ] --- their domain specific usage and how they relate to claims [o9LQ]_

We've added more details about the datasets to the beginning of the empirical studies section: the number of occupations in each dataset, more details about the covariates, and discussion about the consistency of the number of occupations across datasets. To the appendix, we've added more detailed sequence length comparisons across datasets. The empirical studies section now also briefly mentions how the datasets are formed, and links to the section of the appendix that contains more details.

_Whether the pretraining corpus is novel was considered unclear [o9LQ]; authors argued their novelty is using it "in a transfer learning approach to target survey data"_

We've made a few changes to make the transfer learning approach more clear. In the new version of the paper, we use the term "foundation model" to make clear that while CAREER uses resumes, its goal is to model smaller, more representative datasets (we've also changed the title of the paper to emphasize this point). We've also added to the related work section to make clear that while resume datasets have been used for ML applications, our transfer learning strategy, from resumes to surveys, is novel.

_Lack of detail about the modeling approach was criticized in a number of ways: --- how the predicted job-change indicator affects the next-time-step input [o9LQ]; --- how covariates were ultimately used in experiments [o9LQ]; --- description of representations as complex/strong based on architecture alone without analysis of the encodings [o9LQ]; --- lack of formally including the the objective or loss function [o9LQ] --- Similarly, one reviewer wished more detail on how the transfer was performed and how features were extracted from resumes --- One reviewer wishes clearer specification of hyperparameters and the model architecture; authors replied the appendix provides details. Authors stated that a number of details are provided in appendices._

We've added Equation 18 to show explicitly how the job-change indicator can be marginalized out. The beginning of Section 4 now includes more information about how covariates are used in experiments. The end of Section 4 now describes how we assess the complexity of representations (predictive performance for both a task they are trained for and a task they aren't trained for). The formal loss function is now given in Equation 19. The appendix now includes more discussion of hyperparemters and model architecture, and the empirical studies section previews this discussion.

_There was concern about possible posterior collapse and whether regularizations are used [o9LQ]_

We do not observe any posterior collapse, and Appendix G contains information about regularization. We've linked to this section during the discussion of experimental settings in Section 4.

_One reviewer had a concern that the model size with 12 layers might be excessive, and desired discussion of models with fewer parameters [8Vvr]; authors mentioned discussion in the appendix_

We've added more details about the model ablations and hyperparameter search to Appendix G. When discussing the model in Section 4, we state that Appendix G has more details.

---

> ### Author Response · Authors · 2023-12-29
> **Camera-ready version and detailed changes (2/2)**
>
> _To demonstrate advantage of a two-stage model, there was a desire to compare to a one-stage model with pretraining [8Vvr]; authors commented briefly on this._
>
> The new paper now discusses one-stage prediction without pretraining in Appendix E.
>
> _One reviewer wished discussion of other ways to incorporate covariates for example as separate embeddings [8Vvr]_
>
> Appendix G now discusses other approaches for incorporating embeddings that we tried but did not perform as well as the one in the paper.
>
> _The experiments were criticized, including lacking definition of career prediction and forecasting tasks [o9LQ]_
>
> We've added more details to the experiments, including a definition of career prediction and forecasting to the beginning of Section 4.
>
> _Regarding baselines, there was concern whether the CAREER advantage arises more from the transformer architecture than the pretraining approach, and whether other baselines like pretrained representations using NEMO or pretrained-but-not-finetuned CAREER would perform as well [o9LQ]; a baseline transformer model trained directly from the wage data was also desired [bDhb]. One reviewer more generally desired experiments considering different modeling decisions [8Vvr]; and another wished an ablation study [7qrm]. Authors argued in a response that a transferred version of NEMO would not be a fair comparison but provided a comparison to that; authors also provided some results of a non-transferred baseline, and provided some discussion of a baseline with covariates, a baseline of CAREER without finetuning_
>
> We added a pretrained but not finetuned baseline to the wage prediction experiment in the paper, where we found that pretraining was important for CAREER's predictive advantage. We also added more details about ablations to the appendix which we link to in the main text. While CAREER still outperforms NEMO when NEMO is pretrained, we don't believe this is a relevant comparison and have not included it in the modified version of the paper. We added details about the CAREER model that is pretrained but not fine-tuned to Appendix G.
>
> _The analyses were criticized in various ways: --- use of perplexity only vs. also AUROC/ AUPRC or F1 score [o9LQ]; authors provided some AUC results in a response, but the reviewer still felt there was an inconsistency. --- not reporting confidence intervals in Tables 1/2 [o9LQ,bDhb]; authors provided some in a revision --- lacking detail whether NEMO was pretrained using the resume dataset and hence fairness of the comparison [o9LQ]; --- lacking analysis of the attention weights [o9LQ]; authors argued that "the NLP domain has largely moved away from relying on attention-based explanations"_
>
> Inspired by Reviewer o9LQ's suggestion, we added AUC results to the paper. We've also added uncertainty estimates to Tables 1 and 2. When discussing the baselines in Section 4, we added the fact that all baselines were trained only on survey data. Finally, we do not include attention weights because we believe they can be misleading as importance features [1, 2, 3, 4, 5].
>
> [1] Jain and Wallace. Attention is not explanation. 2019
> [2] Serrano and Smith. Is attention interpretable? 2019
> [3] Bastings and Filippova. The elephant in the interpretability room: Why use attention as explanation when we have saliency methods? 2020.
> [4] Brunner et al. On identifiability in transformers. 2019
> [5] Kobayashi et al. Attention is not only a weight: Analyzing transformers with vector norms. 2020

---

> > ### Comment · Action_Editors · 2024-01-22
> > **Thank you for the revisions; I have approved the camera-ready version.**
> >
> > Dear authors, thank you for the revision and the description of changes. I have gone through them and believe they address the reviewers' concerns sufficiently; it is understandable that some proposed additions such as the attention weight analysis can be left out.

---

### Decision · Action_Editor_xQ6M · 2023-11-11

**Recommendation:** Accept with minor revision

**Comment:**

This paper received borderline reviews with three reviewers leaning towards acceptance and one leaning towards rejection. There was significant discussion between authors and one reviewer leaning towards acceptance, but not with the other three reviewers.

Several positive aspects were appreciated:

+ The problem was considered nice [7qrm] and the concrete application of representation and transfer learning was appreciated [bDhb]
+ Some reviewers considered the presentation clear [bDhb] and the paper well-written [8Vvr]
+ The algorithmic approach was considered well-grounded [o9LQ]
+ Incorporation of categorical sequences by initial embedding was considered interesting [o9LQ]
+ Pointing out limitations of claims from resume databases was appreciated [o9LQ]
+ Discussion of prior work was appreciated [o9LQ]
+ The description of the CAREER model construction was considered clear [o9LQ]
+ The experiments [o9LQ] and the shown performance improvements [bDhb,8Vvr,7qrm] were appreciated
+ The training details in the appendix were appreciated [8Vvr]

Many criticisms of the paper were also raised by the reviewers:

- Lack of technical ML innovations was criticized, even considering whether the paper would fit in an applied journal better than TMLR [8Vvr,7qrm]; authors argued the paper fits TMLR's criteria.
- The motivation of the approach was considered insufficient and lacking detail about the data use in the econometric domain, the typical predictive tasks, their importance (including background on direct applications of wage forecasting [bDhb]), and how informative the experiments are for the use case [o9LQ]. Authors mentioned some discussion in a response.
- Lack of detail about data sets was criticized along a number of aspects:
  --- how they are formed, [o9LQ]
  --- numbers of occupations and of covariate values, [o9LQ]
  --- career sequence lengths, [o9LQ]
  --- distribution similarities/differences between pretraining data and survey data sets, [o9LQ]
  --- consistency of the number of occupations across data in the paper [o9LQ]
  --- their domain specific usage and how they relate to claims [o9LQ]
- Whether the pretraining corpus is novel was considered unclear [o9LQ]; authors argued their novelty is using it "in a transfer learning approach to target survey data"
- Lack of detail about the modeling approach was criticized in a number of ways:
  --- how the predicted job-change indicator affects the next-time-step input [o9LQ];
  --- how covariates were ultimately used in experiments [o9LQ];
  --- description of representations as complex/strong based on architecture alone without analysis of the encodings [o9LQ];
  --- lack of formally including the the objective or loss function [o9LQ]
  --- Similarly, one reviewer wished more detail on how the transfer was performed and how features were extracted from resumes
  --- One reviewer wishes clearer specification of hyperparameters and the model architecture; authors replied the appendix provides details.
  Authors stated that a number of details are provided in appendices.
- There was concern about possible posterior collapse and whether regularizations are used [o9LQ]
- One reviewer had a concern that the model size with 12 layers might be excessive, and desired discussion of models with fewer parameters [8Vvr]; authors mentioned discussion in the appendix
- To demonstrate advantage of a two-stage model, there was a desire to compare to a one-stage model with pretraining [8Vvr]; authors commented briefly on this.
- One reviewer wished discussion of other ways to incorporate covariates for example as separate embeddings [8Vvr]
- The experiments were criticized, including lacking definition of career prediction and forecasting tasks [o9LQ]
- Regarding baselines, there was concern whether the CAREER advantage arises more from the transformer architecture than the pretraining approach, and  whether other baselines like pretrained representations using NEMO or pretrained-but-not-finetuned CAREER would perform as well [o9LQ]; a baseline transformer model trained directly from the wage data was also desired [bDhb]. One reviewer more generally desired experiments considering different modeling decisions [8Vvr]; and another wished an ablation study [7qrm]. Authors argued in a response that a transferred version of NEMO would not be a fair comparison but provided a comparison to that; authors also provided some results of a non-transferred baseline, and provided some discussion of a baseline with covariates, a baseline of CAREER without finetuning,
- The analyses were criticized in various ways:
  --- use of perplexity only vs. also AUROC/ AUPRC or F1 score [o9LQ]; authors provided some AUC results in a response, but the reviewer still felt there was an inconsistency.
  --- not reporting confidence intervals in Tables 1/2 [o9LQ,bDhb]; authors provided some in a revision
  --- lacking detail whether NEMO was pretrained using the resume dataset and hence fairness of the comparison [o9LQ];
  --- lacking analysis of the attention weights [o9LQ]; authors argued that "the NLP domain has largely moved away from relying on attention-based explanations"
- Due to lack of various details, it was unclear whether claims were supported by the paper [o9LQ]

Due to the large number of clarity concerns, I suggest to do a minor revision as a clarification pass, to make sure the concerns mentioned in reviews and in author replies are clear enough in the revised paper. Since authors pointed to the appendices in several of their responses, it may be useful to make these links clearer where the information in appendices is central for understanding.

**Audience:**

Overall, reviewers seemed mixed on this aspect, but overall somewhat leaned towards there being interest. Although two reviewers [8Vvr,7qrm] were concerned whether the paper would be better suited elsewhere than TMLR, one of them [8Vvr] did consider the paper a a well written empirical study; and a third reviewer [o9LQ, having a detailed review] did seem to appreciate the technical choices, and a fourth [bDhb] appreciated the paper as a  a very concrete application of representation and transfer learning. Thus, overall the view does seem to be that the application and some technical aspects can be appreciated by some in the TMLR audience.

Thus overall I also lean towards there being at least some individuals in the TMLR audience who would be interested.

**Claims And Evidence:**

The reviewers raised several concerns regarding lack of clarity in the descriptions of the paper, but the authors have fairly comprehensively tried to answer the concerns.

Reviewers also raised several concerns regarding convincingness and clarity of the evidence, in particular desiring several additional comparisons, but authors have now provided additional baselines that seemed to mostly satisfy these points even if some further exploration could have been possible.

Thus, overall it seems the claims are at this stage supported well enough, at least if the authors make sure that the concerns raised by the reviewers are addressed and the answers provided during the review and discussion are thoroughly incorporated.